# Differentiable sorting for censored time-to-event data.

**Andre Vauvelle[12]\*, Benjamin Wild[3]\*, Roland Eils[3], Spiros Denaxas[1]**
University College London[1], BenevolentAI[2], Berlin Institute of Health[3]
{andre.vauvelle.19,s.denaxas}@ucl.ac.uk
{benjamin.wild, roland.eils}@bih-charite.de

## Abstract

Survival analysis is a crucial semi-supervised task in machine learning with significant real-world applications, especially in healthcare. The most common approach to survival analysis, Cox's partial likelihood, can be interpreted as a ranking model optimized on a lower bound of the concordance index. We follow these connections further, with listwise ranking losses that allow for a relaxation of the pairwise independence assumption. Given the inherent transitivity of ranking, we explore differentiable sorting networks as a means to introduce a stronger transitive inductive bias during optimization. Despite their potential, current differentiable sorting methods cannot account for censoring, a crucial aspect of many real-world datasets. We propose a novel method, Diffsurv, to overcome this limitation by extending differentiable sorting methods to handle censored tasks. Diffsurv predicts matrices of *possible* permutations that accommodate the label uncertainty introduced by censored samples. Our experiments reveal that Diffsurv outperforms established baselines in various simulated and real-world risk prediction scenarios. Furthermore, we demonstrate the algorithmic advantages of Diffsurv by presenting a novel method for top-k risk prediction that surpasses current methods. In conclusion, Diffsurv not only provides a novel framework for survival analysis through differentiable sorting, but also significantly impacts real-world applications by improving risk stratification and offering a methodological foundation for developing predictive models in healthcare and beyond.

## 1   Introduction

Survival analysis plays a pivotal role in many realworld machine learning applications, spanning fields such as reliability engineering, marketing, and insurance, with a particularly significant impact in healthcare. The goal of survival analysis is to predict the time until the occurrence of an event of interest, such as death, based on a set of covariates. In clinical studies, these include demographic variables such as sex and age, but may also encompass more complex data modalities such as medical images.

The concept of censoring is a distinguishing characteristic that sets survival analysis apart from conventional machine learning approaches. Particularly prevalent in observational datasets, it refers to situations where event times remain unobserved because a patient might not have undergone the event by the time of data collection. This can be due to a variety of reasons such as the study period ending before all events of interest have occurred or subjects leaving the study.

Overlooking censoring can skew predictions towards the censoring event, rather than the event of interest. This bias becomes particularly noticeable when the study's endpoint can be inferred from the observed covariates - age being a notable example. In such cases, the predicted event times are likely

---

\*Equal contribution

37th Conference on Neural Information Processing Systems (NeurIPS 2023).

to biased towards the censoring event time, thereby neglecting the actual event of interest [Kvamme and Borgan, 2019].

The Cox Proportional Hazards model is widely used for handling censored data in survival analysis [Cox, 1972]. The model optimizes a partial likelihood function over ranked data, considering only the order of events, not their exact time of occurrence. As such, Cox's partial likelihood serves as a ranking loss, learning from the order of patients based on their hazard of experiencing an event, not their exact survival time.

Raykar et al. [2007] showed that Cox's partial likelihood (CPL) and ranking losses can be directly equated, with both providing lower bounds to the concordance index, the primary evaluation metric used in survival analysis. Both losses are foundational to many survival deep learning methodologies like DeepSurv Katzman et al. [2018] and DeepHit Lee et al. [2018].

However, this relation operates under the assumption of pairwise independence. This simplification, while practical, can de-emphasize the transitive properties inherent in survival data. As shown by Goldstein and Langholz [1992], larger risk set sizes can lead to more efficient estimators, suggesting potential benefits in considering listwise ranking losses Cao et al. [2007]. These losses optimize over lists of values rather than individual pairs, thereby better capturing the transitive dynamics of the data. Despite the similarities, listwise losses have remained largely unexplored within the field of survival analysis. This could be partly due to an uncertainty around how to handle censoring.

We propose a new approach that takes advantage of recent developments in continuous relaxations of sorting operations, allowing end-to-end training of neural networks with ordering supervision [Grover et al., 2019, Blondel et al., 2020, Petersen et al., 2021]. This method incorporates a sorting algorithm into the network architecture, where the order of the samples is known, but their exact values are unsupervised. With this, we introduce *Diffsurv*, an extension of differentiable sorting methods that enables end-to-end training of survival ranking models with censored data.

Briefly, our contributions are summarised:

- Our primary contribution is the extension of differentiable sorting methods to account for censoring by introducing the concept of possible permutation matrices. (Section 3.1)

- We empirically demonstrate that our new differentiable sorting method matches or improves risk ranking performance across multiple semi-simulated and real-worlds censored datasets. (Section 4)

- We investigate the role of transitivity in survival analysis and show, through experiments with semi-simulated data, that differentiable sorting networks can benefit from this inherent property of the data. (Section 4)

- We demonstrate that differentiable sorting of censored data enables the development of new methods with practical applications, using the example of end-to-end learning for top-k risk stratification. (Section 3.2)

## 2   Survival Analysis and its Relation to Ranking

A dataset with censored event times is summarized as $\mathcal{D} = \{t_i, \boldsymbol{x}_i, \delta_i\}_{i=1}^N$, where $N$ is the total number of patients. For a patient $i$, the time-to-event $t_i$ is the minimum of the true survival time $t_i^*$ and the censoring time $c_i^*$, with $\delta_i$ indicating whether an event ($t_i^* \leq c_i^*$, $\delta_i = 1$) or censoring ($t_i^* > c_i^*$, $\delta_i = 0$) was observed. Covariates are $\boldsymbol{x}_i \in \mathbb{R}^d$ representing a $d$-dimensional vector but the methods discussed here also generalise to higher dimensional tensors such as image data.

The widely-used method for addressing censoring in survival analysis is the Cox Partial Likelihood (CPL) model, introduced by Cox [1972]. The CPL is designed to maximize the following general form:

$$\mathcal{L}(\theta) := \prod_{i:\delta_i=1} \frac{f_\theta(\boldsymbol{x}_i)}{\sum_{j:t_j>t_i} f_\theta(\boldsymbol{x}_j)}, \tag{1}$$

Figure 1: Differentiable Sorting for Censored Time-to-Event Data. Inputs, in this case SVHN images, are transformed into scalar values through a neural network. A differentiable permutation matrix, $P$, is computed using sorting networks. The model can be optimized for downstream tasks, such as risk stratification and top-k highest risk prediction, by using the matrix $Q_p$ of possible permutations based on the observed events and censoring.

where $f_\theta$ is the hazard function, a score prediction function estimating the probability of an event at a particular time, given input features $x_i$. The product only includes uncensored patients, whereas the denominator term also includes censored patients with $t_j > t_i$.

Reflecting the structure of survival data, the Cox Partial Likelihood (CPL) model compares individuals still "at risk" at each time point, similar to a nested case-control study. This directly shapes the likelihood equation in CPL, with the numerator representing the hazard function for the event-experiencing individual, and the denominator summing over all individuals still at risk.

Extensions of the Cox model, like [Katzman et al., 2018] and [Kvamme et al., 2019], have modified $f_\theta = \exp(\theta \cdot x_i)$ to relax the linear covariate interaction and proportional hazards assumptions. Introducing neural networks $h_\theta$ to handle the non-linearity, adjusting $f_\theta$ to be $f_\theta = \exp(h_\theta(x_i))$, and to manage non-proportional hazards, they set $f_\theta = \exp(h_\theta(x_i, t_i))$.

## 2.1 Pairwise Independence

Both of these previous works note that the risk set $\mathcal{R} = \{j : t_j > t_i\}$ is intractable for deep learning applications as it considers all comparable patients. To mitigate memory constraints, we can sample a fixed-size risk set, denoted as $\tilde{\mathcal{R}}$, such that $|\tilde{\mathcal{R}}| = n < N$. Kvamme et al. [2019] go further, arguing it is reasonable to take a constant sample size of 1 and include the individual $i$ in the risk set (such that $n = 2$). This leads to the simplified loss of the form

$$\mathcal{L}(\theta) = \prod_{i:\delta_i=1} \frac{f_\theta(x_i)}{f_\theta(x_i) + f_\theta(x_{J(i)})}, \text{ where } J(i) \in \mathcal{R} \setminus \{i\}. \tag{2}$$

Further, take the mean log partial likelihood to be

$$\text{loss} = \frac{1}{n_e} \sum_{i:\delta_i=1} \log(1 + \exp[h_\theta(x_{J(i)}) - h_\theta(x_i)]), \text{ where } J(i) \in \mathcal{R} \setminus \{i\}, \tag{3}$$

where $n_e$ is the number of non-censored events. In this simplified form, it can be seen that the partial likelihood only considers the pairwise relative ordering or ranking of survival times.

The concordance index or c-index Harrell et al. [1982] is a commonly used as an evaluation for survival analysis methods and is a generalization of the Area Under the Receiver Operating Characteristic Curve (AUROC) that handles right-censored data. It is defined as

$$\text{c-index} := \frac{1}{n} \sum_{i:\delta_i=1} \mathbb{1}(f(x_i) < f(x_j)), j \in \mathcal{R} \setminus \{i\}. \tag{4}$$

Raykar et al. [2007] first showed that the Cox's partial likelihood is approximately equivalent to maximizing the concordance index or c-index and that closer bounds can be found by minimizing the

general ranking loss, with acceptable pairs $\mathcal{A} = \{(i,j) : \delta_i = 1 \wedge t_j > t_i\}$ and

$$\text{ranking-loss} := \frac{1}{|\mathcal{A}|} \sum_{(i,j) \in \mathcal{A}} \phi(f_\theta(\boldsymbol{x}_i) - f_\theta(\boldsymbol{x}_j)), \tag{5}$$

where $\phi$ is a function that relaxes the non-differentiable $\mathbb{1}$ of the c-index. From Equation 3, it can be seen that $\phi : x \rightarrow -\log(1+\exp(-x)) = \log(\sigma(x))$. Here, we have shown that the simplifications to the partial likelihood made by Kvamme et al. [2019] are equivalent to using the log-sigmoid ranking loss.

The key difference between ranking and partial likelihood losses comes when considering the assumption that it is reasonable to take a constant sample size of 2 (one pair in the risk set) in the partial likelihood. This effectively introduces the assumption that each pair (i, j) is independent of any other pair. However, this assumption seems puzzling given the inherent transitivity of ranking (if $i > j$ and $j > k$ then $i > k$).

## 2.2 Listwise Ranking and Differentiable Sorting Networks

[Cao et al., 2007] first proposed the notion of a *listwise* ranking loss, arguing its benefits in situations where the entire order of items is of importance. This approach treats the ranking problem as a permutation problem, with the aim of learning a model that can provide the optimal permutation of an entire list of items, rather than considering independent pairs. Indeed, this idea is closely related to the partial likelihood from Cox's original work. As pointed out by Wang and Yang [2022], the Top-1 probability method originally proposed by Cao et al. [2007] and extended to ListMLE by Xia et al. [2008] takes the same form as CPL.

More recent works closely related to listwise ranking have begun to explore combining traditional sorting algorithms with differentiable sorting functions [Petersen et al., 2021]. Sorting networks are a family of sorting algorithms that consist of two basic components: wires and conditional swaps [Batcher, 1968, Knuth, 1998]. Wires carry values to be compared at conditional swaps. If one value is bigger than the other then the values carried forward are swapped around. This allows construction of *sorting networks* that can provably sort a list of values. Conditional swaps are exactly the min and max operators that ensure that with inputs $\{a, b\}$ and outputs $a^* \leq b^*$, $a^* = \min(a, b)$ and $b^* = \max(a, b)$. Examples of odd even and bitonic networks are shown in Appendix C.

In order to train models based on ordering information alone, differences between predicted and true orderings must be backpropagated through sorting algorithms. However, they often require the use of non-differentiable $\max$ and $\min$ operators. These are analogous to the non-differentiable indicator function that was discussed earlier in the c-index equation 4. Differentiable sorting methods, similar to ranking losses, rely on approximating these operators with smooth alternatives [Grover et al., 2019].

Petersen et al. [2021] propose combining traditional sorting networks and differentiable sorting functions. Consider that when $a$ asymptotically approaches $b$, the transition point where it surpasses $b$ is non-continuous and therefore non-differentiable. Just as previously shown in the ranking loss, such operations can be made differentiable using the logistic relaxation

$$\min_\sigma(a, b) = a \cdot \sigma(b - a) + b \cdot \sigma(a - b) \text{ and } \max_\sigma(a, b) = a \cdot \sigma(a - b) + b \cdot \sigma(b - a). \tag{6}$$

If an inverse temperature parameter $\beta > 0$ is introduced such that $\sigma : x \rightarrow \frac{1}{1+e^{-\beta x}}$, then as $\beta \rightarrow \infty$ the functions tend to the exact $\min$ and $\max$ functions. Other relaxations of the step function can also be considered, Petersen et al. [2022a] show that the Cauchy distribution preserves monotonicity which is desirable for optimization. Given this, we use the Cauchy distribution as our relaxation for all experiments, where $\sigma : x \rightarrow \frac{1}{\pi}\arctan(\beta x) + \frac{1}{2}$.

For an input list to be ordered, each layer of the sorting network can be considered an independent permutation matrix $\boldsymbol{P}_l$ with elements given by

$$P_{l,ii} = P_{l,jj} = \sigma(a_j - a_i) \text{ and } P_{l,ij} = P_{l,ji} = 1 - \sigma(a_j - a_i), \tag{7}$$

where $a$ signifies intermediate values being compared. The first layer is input with $z_i = h_\theta(\boldsymbol{x}_i)$, each vector of covariates or images being processed independently by the same neural network. The

indices being compared at each layer are determined by the sorting network and the final predicted probability matrix is the product of each layer of sorting operations,

$$P = \left( \prod_{l=1}^{n} P_l^{\mathsf{T}} \right)^{\mathsf{T}}.$$ (8)

Where $P$, is the final doubly-stochastic permutation matrix, doubly-stochastic meaning that the rows and columns both sum to 1. It is possible to interpret each element $P_{ij}$ of the predicted permutation matrix as the predicted probability of permuting from a randomly assigned rank $i$ to a true rank $j$. Finally, we can define a loss by minimizing the cross-entropy between the ground truth orders represented by true permutation matrix $Q$ and predicted permutation matrix $P$ as

$$\mathcal{L} := \sum_{c=1}^{n} \left( \frac{1}{n} \text{CrossEntropy}(P_c, Q_c) \right),$$ (9)

where $P_c$ and $Q_c$ denote the c-th columns of their respective matrices.

## 2.3 Differentiable Sorting Networks Relation to Ranking and Partial Likelihood

It is possible to directly relate differentiable sorting networks with ranking losses and partial likelihood. Expanding out the cross entropy loss, we find

$$\mathcal{L} = \sum_{c=1}^{n} \left( \frac{1}{n} \sum_{i=1}^{n} q_{ic} \log(p_{ic}) \right),$$ (10)

where $q_{ic} = 1$ only when $i$ is the true rank otherwise 0. Each $p_{ic}$ is always a function of the difference in pairs of inputs $x_i$ and $x_j$. This is complicated by the products of intermediate values $a$ introduced by the sorting network but denoted as

$$p_{ic} = \prod_{(a_i, a_j) \in \mathcal{P}_l : l=1}^{n} \sigma(a_i - a_j)$$ (11)

where $\mathcal{P}_l$ to denotes the set of comparisons to be made at each layer of the sorting network. With $n = 2$ and $\beta = 1$, a sorting network only requires a single relaxed conditional swap and the loss returns to the same recognisable log-sigmoid ranking loss in Equation 5, and Cox negative log partial likelihood in Equation 3.

# 3 Methods

## 3.1 Diffsurv: Handling Censoring with Possible Permutation Matrices

For risk sets of size 2, given proper case-control sampling, it will always be possible to define a single ground truth permutation matrix $Q$. However, when venturing to higher risk set sizes, differentiable sorting methods can no longer handle censoring since there is not a single ground truth permutation matrix $Q$. We cannot determine the exact rank of patients who are censored before another who experienced an event. It is only possible to know the range of possible ranks to which a patient should belong. In Figure 2, we provide an illustration demonstrating the possible ranks for a number of censored and uncensored events.

Though we no longer have access to a single permutation matrix, we may instead consider the set of all possible permutation matrices, $\mathcal{Q} = \{Q_1, Q_1, \ldots, Q_\kappa\}$. In the best case, all values are uncensored and $|\mathcal{Q}| = 1$ and in the worse case, when all patients are censored $|\mathcal{Q}| = n!$. Our primary contribution is to extend differentiable sorting methods to censored ranks by discriminating between possible and impossible permutations.

We introduce a more computationally tractable representation of $\mathcal{Q}$ by defining the *possible permutation matrix*, $Q_p$, which is the element-wise maximum of every permutation in $\mathcal{Q}$,

$$Q_{pij} = \max\{Q_{1ij}, Q_{2ij}, \ldots, Q_{\kappa ij}\}.$$ (12)

For survival analysis, it is possible to determine $Q_p$ in linear time given a sorted list of event times $t_i$ and event indicators $\delta_i$. We will consider higher ranks to correspond with a smaller time-to-event. Let us consider two scenarios:

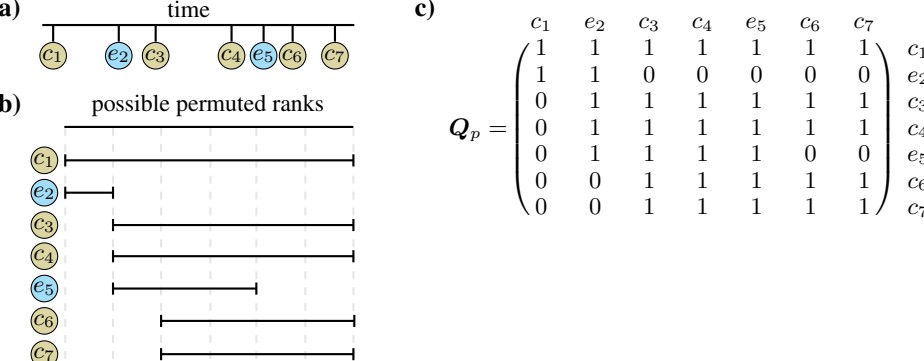

Figure 2: For an example case (**a**) with two events (○, $e_1$ and $e_5$) and multiple censored samples (○, $c_1, c_3, c_4, c_6, c_7$) the uncertainty in the possible permuted rankings (**b**) due to censoring is taken into account to derive the possible permutation matrix $\boldsymbol{Q}_p$ (**c**).

1. For a right-censored individual $i$ (i.e., $\delta_i = 0$), the possible ranks must be lower than the ranks of preceding uncensored events. We can express the set of possible ranks $\mathcal{R}_i$ as:

$$\mathcal{R}_i = \{r \mid r < \text{rank}(j), \forall j : \delta_j = 1, t_j < t_i\} \tag{13}$$

This set includes all ranks $r$ that are less than the rank of any uncensored patient $j$ with an event time $t_j$ preceding the censoring time $t_i$ of the individual $i$.

2. For an uncensored individual $i$ (i.e., $\delta_i = 1$), the possible rank must be lower than all preceding uncensored events and higher than the ranks of all subsequent events. We can define the set of possible ranks as:

$$\mathcal{R}_i = \{r \mid r < \text{rank}(j), \forall j : \delta_j = 1, t_j < t_i\} \cap \{r \mid r > \text{rank}(j), \forall j : t_j > t_i\} \tag{14}$$

This set includes all ranks $r$ that satisfy both conditions: being less than the rank of any preceding uncensored patient $j$ with $t_j < t_i$ and greater than the rank of any subsequent patient $j$ with $t_j > t_i$.

With these observations, it's straightforward to construct $\boldsymbol{Q}_p$. If it's feasible for patient $i$ to permute to rank $j$, i.e. $j \in \mathcal{R}_i$, then $\boldsymbol{Q}_{pij} = 1$, otherwise $\boldsymbol{Q}_{pij} = 0$. See Figure 2 (c) for a visual representation of $\boldsymbol{Q}_p$.

Given the possible permutation matrix $\boldsymbol{Q}_p$ and the predicted permutation matrix $\boldsymbol{P}$, the vector of probabilities $\boldsymbol{p}$ of a value being ranked within the set of possible ranks can be computed. Although the ground truth probabilities are unknown, the range of possible ranks is known, and the model can be optimized to maximize the sum of the predicted probabilities of all possible ranks for each sample. Noted here as the column-sum of the element-wise product $\circ$, between $\boldsymbol{Q}_p$ and $\boldsymbol{P}$.

$$\boldsymbol{p} = \sum_{j=1}^{n} (\mathbf{Q}_p \circ \mathbf{P})_{i,j}. \tag{15}$$

The binary cross-entropy loss can then be easily applied

$$\mathcal{L} = \sum_{i=1}^{n} -y_i \log(p_i) - (1 - y_i) \log(1 - p_i) \tag{16}$$

where $y_i$ indicates whether set of predicted ranks is possible or impossible.

Equation 15 accounts for the potential challenges of incorporating right-censored samples. The binary cross-entropy remains identical whether the model predicts uniform probability for all possible ranks or concentrates the probability mass on a single rank possible rank.

The introduction of the possible permutation matrix can be used in conjunction with any differentiable sorting method that outputs a doubly-stochastic permutation matrix. This includes methods such

as SinkhornSort from Cuturi et al. [2019]. Though, in this paper, we will restrict our focus to the discussed differentiable sorting networks. We refer to the use of differentiable sorting networks and the possible permutation matrix as *Diffsurv*.

## 3.2 Top-K risk prediction

Finally, we demonstrate how the algorithmic supervision of sorting algorithms enables the development of novel methods in survival analysis, using the example of top-k risk prediction. In practical settings, it is often not necessary to rank all samples correctly. Rather, it is essential to identify the samples with the highest risk, such as by a healthcare provider, to prioritize care and interventions.

With Diffsurv, top-k risk prediction is straightforward to implement by modifying the loss such that the negative log-likelihood of predicted top-k ranks in $\boldsymbol{P}$ is maximised for individuals with a possible permutation to *any* top-k rank according to $\boldsymbol{Q}_p$.

First, let's denote $\mathcal{T}_k$ as the set of values with a possible permutation to a top-k rank, derived from the ground truth possible permutation matrix $\boldsymbol{Q}_p$:

$$\mathcal{T}_k = \{i | \sum_{j=1}^{k} \boldsymbol{Q}_{pij} > 0\} \tag{17}$$

Importantly, due to the uncertainty introduced by censoring, the set of individuals with a possible permutation to a top k rank $\mathcal{T}_k$ can be arbitrarily large. For example, in case all individuals are censored, $\mathcal{T}_k$ is the set of all individuals. Then, the top-k loss is described as:

$$\mathcal{L}_{\text{top-k}} = - \sum_{i \in \mathcal{T}_k} \log \left( \sum_{j=1}^{k} \boldsymbol{P}_{ij} \right). \tag{18}$$

This loss is minimized when the model correctly predicts a top-k rank for the indices in $\mathcal{T}_k$. This represents the individuals with possible permutations to the top-k highest risk ranks. Importantly, this loss function is optimized for the identification of potential top-k high-risk individuals, without considering the specific order within these top-k ranks. To establish a baseline for comparison with Diffsurv's top-k risk prediction, we also introduce two variants of the Cox Partial Likelihood method. In the first variant, we adjust the likelihood term so that the product considers only the set of patients who have a potential permutation to a top-k rank, according to the matrix of possible permutations $\boldsymbol{Q}_p$:

$$\mathcal{L}_{\text{CPL\_I}} = \prod_{i:i \in \mathcal{T}_k} \frac{f_\theta(\boldsymbol{x}_i)}{\sum_{j:t_j > t_i} f_\theta(\boldsymbol{x}_j)} \tag{19}$$

In the second variant, we further limit the set of patients to those who have both experienced an event and have a possible permutation to a top-k rank:

$$\mathcal{L}_{\text{CPL\_II}} = \prod_{i:\delta_i=1 \wedge i \in \mathcal{T}_k} \frac{f_\theta(\boldsymbol{x}_i)}{\sum_{j:T_j > T_i} f_\theta(\boldsymbol{x}_j)}. \tag{20}$$

Note that the denominator term is unchanged in both variants and considers only comparable pairs and includes censored patients $T_j > Ti$. Evaluation of top-k risk prediction is also complicated by the uncertainty due to censoring. For Diffsurv and both variants of the Cox Partial Likelihood, we can first define the set of individuals predicted to be within the top-k highest risk:

$$\mathcal{P}_k = \{i | \text{rank}(f_\theta(x_i)) \geq k\} \tag{21}$$

We can then define the fraction of how many of these individuals are in the set of possible top-k highest ranks $\mathcal{T}_k$ to evaluate the top-k risk prediction performance:

$$\text{top-k-score} = \frac{|\mathcal{P}_k \cap \mathcal{T}_k|}{|\mathcal{P}_k|} \tag{22}$$

## 4 Experiments

In our experiments, we aim to assess the performance of *Diffsurv* and compare it against the conventional Cox Partial Likelihood (CPL) methods. Initially, we focus on confirming the importance

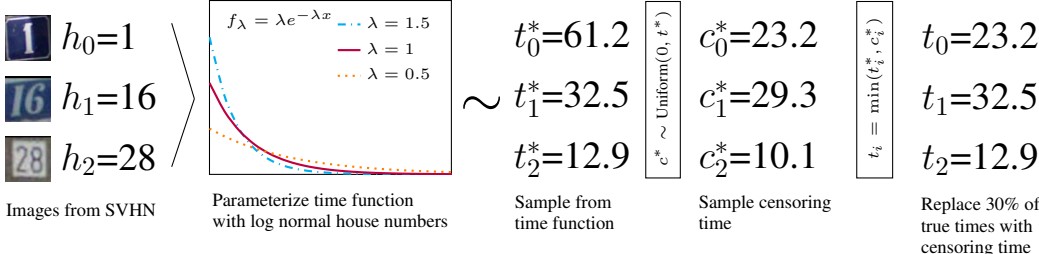

Figure 3: Visual abstract of the survSVHN dataset.

of taking a listwise approach and evaluating the ability of differentiable sorting networks to better capture the inherent transitivity in semi-simulated data. Subsequently, we extend our analysis to compare Diffsurv and its top-k extension across multiple publicly available real-world datasets.

**Baselines**: We compare *Diffsurv* primarily against Cox's Partial Likelihood, using the Ranked List implementation from pycox [Kvamme et al., 2019]. We include Efron and Breslow estimates of CPL from Yang et al. [2022] for survSVHN. For smaller datasets, we add non-deep learning baselines: Lifelines' Cox Regression [Davidson-Pilon, 2019] and sksurv's Random Survival Forests [Pölsterl, 2020]. We do not compare with DeepHit [Lee et al., 2018] since we do not model non-proportional hazards. For an extended discussion, see Appendix A.

**Network Architectures**: For both CPL and Diffsurv, we use a fixed neural network architecture depending on the dataset. Small datasets utilize a single-layer neural network, survSVHN uses a ConvNet architecture as in Petersen et al. [2021], and MIMIC IV CXR uses EfficientNet-B0 [Tan and Le, 2020].

**Training and Evaluation**: We employ AdamW for optimization. Validation approach varies: for smaller datasets, we apply nested 5-fold cross-validation, while for imaging datasets we use train:val:test splits. We performed hyperparameter tuning for learning rate, weight decay, batch size, and risk set size. In the case of imaging datasets, we maintained fixed values for learning rate and weight decay. As in Petersen et al. [2021], we determine steepness as a function of the risk set size $n$, $\beta = 2n$ for odd-even and $\beta = (\log_2 n)(1 + \log_2 n)$ for bitonic. The type of sorting network can either be bitonic or odd-even and is determined during hyperparameter tuning. Further details on the experimental setup, including compute time, are provided in Appendix B and at https://github.com/andre-vauvelle/diffsurv.

**Semi-synthetic survSVHN**: Based on the Street View House Numbers (SVHN) dataset [Netzer et al., 2011], we simulate survival times akin to survMNIST Pölsterl [2019]. The increased complexity of SVHN over MNIST offers a testbed which is better able to discern the performance differences between methods. Each house number parameterizes a beta-exponential time function for survival times. Risk parameters or hazards $\lambda_i$ are calculated as the logarithm of house numbers, standardized and scaled for a mean survival time of 30. We introduce censoring by randomly selecting 30% of house numbers and replacing true times with values sampled uniformly between $(0, t_i]$ (See Figure 3).

We can examine the implications of inherent transitivity within the data. Instead of parameterizing a time function based on unique hazards derived from house numbers, we group $\lambda_i$ into distinct hazard quantiles. Each quantile encompasses a set of house numbers associated with a similar hazard level. We then calculate the transitivity ratio, defined as $\frac{\text{\# of transitive triplets}}{\text{\# of triplets}}$, where a sampled triplet is considered transitive if $(\lambda_i > \lambda_j > \lambda_k)$.

This methodology provides us with a means to control the degree of transitivity in our data. At one extreme, we might categorize data into only two groups, representing the lower and upper halves of house numbers, which results in a transitivity ratio of 0. At the other extreme, each house number could constitute its own unique category (indicated by $\infty$), leading to high transitivity.

Our results, summarized in Table 1, align with the expectations laid out in Section 2.3: both Diffsurv and CPL methods perform similarly when the risk set size is at its minimum ($n = 2$). However, with the expansion of the risk set size, the performance of the two methods diverges, with Diffsurv consistently outperforming CPL. Table 2 sheds light on a potential reason for this divergence. As

Table 1: Results for training on survSVHN with increasing risk set size. Mean (standard deviation) c-index over 5 trails with different seeds. $^\dagger$ When $n = 2$ both methods are equivalent to the ranking loss to up continuous relaxation of swap operation.

| Risk set size | $2^\dagger$ | 4 | 8 | 16 | 32 |
|---|---|---|---|---|---|
| Diffsurv | .918 (.003) | **.934** (.002) | **.940** (.001) | **.943** (.002) | **.941** (.002) |
| Cox Partial Likelihood | .913 (.002) | .925 (.002) | .931 (.002) | .933 (.002) | .930 (.003) |

Table 2: Results for training on survSVHN while increasing transitivity. Metric is c-index. Mean performance over 3 trails with different seeds. **Bold** indicates significant improvement (t-test, $p \le 0.01$). Restricted to a fixed batch size and risk set size of 32.

| Number of Quantiles | 2 | 4 | 8 | 16 | 32 | 64 | 128 | $\infty$ |
| Transitivity Ratio | .0 | .374 | .657 | .819 | .908 | .954 | .975 | .991 |
|---|---|---|---|---|---|---|---|---|
| Diffsurv: Bitonic | .643 | .803 | .882 | .922 | .933 | .939 | .939 | .939 |
| Diffsurv: Odd Even | .646 | .802 | .883 | **.923** | **.935** | **.939** | **.940** | **.941** |
| CPL: Ranked List | .651 | .803 | .880 | .909 | .916 | .920 | .921 | .920 |
| CPL: Efron | .647 | .801 | .871 | .898 | .904 | .905 | .909 | .908 |
| CPL: Breslow | .648 | .801 | .871 | .898 | .904 | .909 | .907 | .910 |

the number of quantiles is increased, thereby enhancing the degree of transitivity within the data, Diffsurv-based methods start to surpass CPL methods. This finding underscores the role of transitivity in survival data and validates Diffsurv's effectiveness in encapsulating this inherent property. Despite the strong performance of Diffsurv, the C-index for the ground truth risks is 0.980, which is still far above 0.943 for Diffsurv, highlighting the challenging nature of the survSVHN dataset.

**Real-world datasets**: We assess our methods on several public datasets: Four small, popular real-world survival datasets (FLCHAIN, NWTCO, SUPPORT, METABRIC) [Kvamme et al., 2019] and the MIMIC IV Chest X-Ray dataset (CXR) with death as the event [Johnson et al., 2019]. Further details in Appendix B.1.

The results presented in Table 3 demonstrate that Diffsurv achieves equal to or better performance on all datasets analyzed. Additionally, when Diffsurv is optimized for predicting the top 10% of highest-risk individuals, it matches or outperforms Cox's partial likelihood on the real-world datasets.

## 5 Conclusion

Diffsurv introduces a new perspective in survival analysis with censored data, highlighting the relations between survival analysis and the listwise ranking. Our experiments show the effectiveness of differentiable sorting methods for improving survival analysis predictions. Notably, Diffsurv matches or surpasses the performance of the established CPL methods across all examined datasets.

Crucially, Diffsurv sheds light on the importance of transitivity in ranking and survival data, revealing that methods sensitive to this inherent property, such as Diffsurv, show improved performance over those that are not. This insight underscores the value of a listwise approach in dealing with survival data and encourages further exploration for methods that promote a transitive inductive bias.

Moreover, Diffsurv provides a foundation for the development of innovative methods, including the top-k risk stratification method introduced in this work. Beyond survival analysis, the introduction of the possible permutations carries potential for other tasks that involve ranking based on limited order information. The utilization of specialized sorting networks, such as splitter selection networks as in Petersen et al. [2022b], could further leverage partial order information.

Though promising, this work is not without limitations. Future research could focus on extending its applicability to non-proportional hazards and understanding the impact of ties. Moreover, investigating how well it can recover survival functions using approaches like Breslow's estimator and evaluating with Brier scores would provide valuable insights into its potential and limitations.

Table 3: Results for real-world and semi-synthetic datasets. Mean (standard deviation). Survival metric is c-index, Top 10% metric is top-k-score. **Bold** indicates significant improvement (t-test, $p \leq 0.01$).

| | FLCHAIN | NWTCO | SUPPORT | METABRIC | MIMIC IV CXR | survSVHN |
|---|---|---|---|---|---|---|
| Size | 6,524 | 4,028 | 8,873 | 1,904 | 377,110 | 248,823 |
| Censored Proportion | 69.9% | 85.8% | 32.0% | 42.1% | 60.9% | 30.0% |
| **Survival** | | | | | | |
| Cox Regression | .750 (.083) | .692 (.021) | .598 (.010) | .628 (.013) | - | - |
| Random Survival Forest | .789 (.011) | .691 (.024) | .614 (.009) | .641 (.012) | - | - |
| Cox Partial Likelihood | .794 (.013) | .709 (.015) | .642 (.006) | .698 (.011) | .760 (.002) | .933 (.002) |
| Diffsurv | .793 (.009) | .703 (.026) | .645 (.002) | .684 (.011) | **.763** (.001) | **.943** (.002) |
| **Top 10% prediction** | | | | | | |
| Cox Partial Likelihood | .460 (.013) | .390 (.068) | .280 (.023) | .249 (.065) | .390 (.010) | - |
| CPL-TopK (Variant I) | .469 (.007) | .413 (.061) | .479 (.016) | .527 (.083) | .408 (.008) | - |
| CPL-TopK (Variant II) | .460 (.009) | .413 (.054) | .479 (.035) | .487 (.058) | .406 (.006) | - |
| Diffsurv | .452 (.011) | .395 (.082) | .296 (.015) | .331 (.102) | .412 (.002) | - |
| Diffsurv-TopK | **.482** (.019) | **.421** (.065) | **.508** (.027) | .533 (.092) | .412 (.009) | - |

Furthermore, it is important to note that Diffsurv is a survival ranking method and thus can not be used to directly estimate the expected duration until an event occurs.

Overall, Diffsurv constitutes a meaningful advancement in survival analysis, showcasing its significant potential for enhancing risk prediction in real-world use cases. It not only demonstrates promising performance improvements, but also introduces new directions for future research, thereby making a valuable contribution to the field.

# 6 Acknowledgements

We extend our gratitude to Felix Peterson, Leon Sixt, and Samuel Holt for their insightful discussions and invaluable feedback, which significantly contributed to the quality of this work. We also thank the anonymous reviewers for their thoughtful feedback and fruitful discussions, which have been instrumental in enhancing the manuscript.

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

# A Non-proportional Hazards

Our current implementation of Diffsurv operates under the proportional hazards assumption. While this may not fully capture the intricacies of some survival analysis problems—particularly those involving non-proportional hazards—it does not necessarily limit the model's effectiveness in scenarios where the goal is to assess cumulative risk from a fixed index date or from the date of an imaging study. This aligns with the necessity of time-dependent modifications to the C-index for non-proportional models as indicated by Antolini et al. [2005].

If we are primarily interested in understanding the cumulative hazard of an event occurring rather than tracking changes in the hazard over time, the assumption of proportional hazards becomes less pivotal. As such, Diffsurv and CPL remain valuable tools for these cases.

Despite the current limitation of Diffsurv to proportional hazards, it is conceivable that an extension to accommodate non-proportional hazards could be developed, similar to adaptations made for the CPL method.

For instance, as we briefly mentioned earlier, continuous-time extensions of partial likelihood can be used to enable non-proportional hazards Kvamme et al. [2019]. Implemented by directly modeling temporal covariates as $f_\theta = \exp(h_\theta(x_i, T_i))$.

Another class of methods focuses on discretizing the time-to-event variable and modeling the probability mass function (PMF) of event times. For instance, the DeepHit model Lee et al. [2018] employs a neural network architecture to learn the relationships between input features and discretized time-to-event outcomes. Time discretization facilitates modeling of non-proportional hazards but introduces two significant challenges: 1) sensitivity to the choice of time intervals, which can affect the model's accuracy and interpretability, and 2) increased computational complexity, as predictions must be made for each time interval. These models can be computationally expensive, especially for deep learning-based models like DeepHit, making them less suitable for high-dimensional and large-scale datasets, such as the imaging dataset used in this study.

Several future work proposals arise from these observations. First, differentiable sorting could explore the approach of directly modeling temporal covariates, resulting in a time-parameterized predicted permutation matrix. Second, extending Diffsurv to discrete time could be achieved by parameterizing a predicted permutation matrix for each time discretization.

# B Training and evaluation

## B.1 Datasets and Preprocessing

As in Goldstein and Langholz [1992] and Kvamme et al. [2019], we ensure that each risk set contains a valid risk set by sampling controls for a given case. Each batch consists of a number of risk sets such that the input data has shape (batch size, risk set size, covariate shape).

We provide an additional description of each small realworld dataset:

- **FLCHAIN dataset:** A dataset containing information on patients with monoclonal gammopathy of undetermined significance (MGUS), focusing on serum free light chain (FLC) levels to study their prognostic significance in predicting disease progression. Number of covariates: 8.

- **NWTCO dataset:** A dataset from a series of clinical trials on the treatment and outcomes of children with Wilms' tumor, a type of kidney cancer, aiming to improve understanding of tumor biology and optimize treatment strategies. Number of covariates: 9.

- **SUPPORT dataset:** A dataset from a multi-center study investigating the prognosis and treatment preferences of seriously ill hospitalized adults, with the goal of improving end-of-life care and informing decision-making processes. Number of covariates: 22.

- **METABRIC dataset:** A dataset comprising genomic and clinical data on breast cancer patients, focused on uncovering novel molecular subtypes for more precise prognostication and personalized treatment strategies. Number of covariates: 9.

Covariate preprocessing follows [Kvamme et al., 2019], and includes standardising continuous variables and one-hot encoding categorical variables.

**MIMIC IV CXR**: For the survival task, we extract death events from the MIMIC IV dataset. This is done by merging the data on "subject_id", "study_id", and "dicom_id" with the patient table from MIMIC IV, and on "subject_id" with the admission table. For patients without a recorded date of death, censoring dates are be determined as 1 year after the last recorded discharge date for each patient. We exclude 29,345 images without any matches in the MIMIC IV patient table, 19,337 images taken after the latest found discharge date and 55 images taken after a recorded date of death. Time to event is calculated as the number of days from the image study date to either date of death or the censoring date. For MIMIC IV CXR, images undergo several standard transformations: a random horizontal flip and a 15-degree rotation, resizing to 230 x 230 pixels, a 224 x 224 pixel center crop, and conversion to grayscale with three output channels. The data is then transformed into tensors and normalized using ImageNet's mean and standard deviation values. Finally, the train:val:test split of 8:1:1 is done at the patient level ensuring no images from a patient in the test set was found in the training data.

**survSVHN**: In this semi-synthetic dataset, we sample survival times using the beta-exponential distribution. The beta distribution used to sample from the exponential uses a fixed value of 500 for both shape parameters. We follow Petersen et al. [2021] by cropping the centered multi-digit numbers with a boundary of 30%, resizing it to a resolution of 64×64, and then selecting 54 × 54 pixels at a random location. For survSVHN the train:val:test split is provided by Netzer et al. [2011] as is 230,755:5,000:13,068.

## B.2 Model Architecture and Hyperparameters

For the smaller real-world datasets, the hazard function $f_\theta$ is a small fixed Multi-layer Perceptron network with 1 hidden layer and 64 hidden nodes. We also apply a fixed dropout rate of 0.1. Learning rate, weight decay, batch set size and risk set size were found using a grid search across the possible values in Table 4.

Table 4: Hyperparameter values for small real-world datasets.

| Hyperparameter | Values |
| --- | --- |
| Learning rate | [0.1, 0.01, 0.001, 1e-4] |
| Weight decay | [0.1, 0.01, 0.001, 1e-4, 1e-5, 0] |
| (Batch size, risk set size) | [(32, 8), (16, 16), (8, 32), (4, 64), (1, 256)] |

For imaging datasets, we fix learning rate and weight decay for both CPL and Diffsurv. For both survSVHN and MIMIC IV CXR, we use a fixed learning rate of $10^{-4}$ and weight decay of $10^{-5}$. We also used early stopping with a patience of 20 epochs and a maximum of 100,000 training steps.

**survSVHN**: As per Petersen et al. [2021], the model consists of four convolutional layers (with a 5x5 kernel size and 32, 64, 128, 256 filters), each followed by ReLU and max-pooling (2x2 stride). The architecture concludes with a fully connected layer of 64 units, another ReLU, and a one-unit output layer.

**MIMIC IV CXR**: Here, we use EfficientNet-B0 with an added linear layer for single output. We first train the linear prediction layer alone for the initial 2,000 steps. After this, we continue training, this time including both the EfficientNet-B0 and the linear prediction layer.

For the results in Table 1, we keep a fixed batch size of 100. We also provide a comparison where the number of values is fixed in each batch in Table 9.

Note that during evaluation, the sorting network is not used since we only need to evaluate the ranks of the trained risk scores. Similarly, case-control sampling is not used. We measure the ranking performance of the models using the concordance index.

Further implementation details and the best hyperparameters for each dataset are provided at `anon@ git.com`.

## B.3 Compute Requirements

Experiments on smaller real-world datasets are compact enough to facilitate effective training on a CPU, with each variant, including the CPH baselines, completing per experiment in less than 20 minutes. However, the larger imaging datasets require more significant computational power. In the most demanding case, the MIMIC IV CXR experiments, run on an 11GB NVIDIA GeForce GTX 1080 Ti, took roughly 18.5 hours per experiment. Both Diffsurv and CPH methods exhibited comparable run times; however, the Bitonic variant was the fastest, with a lead of approximately 6 minutes. All neural network baselines were implemented using PyTorch and PyTorch Lightning. Although measures were taken to reduce compute time and complexity, such as using half-precision and distributed data parallel (DDP) training strategies, the overall training times are far from optimized.

To understand the runtime differences of Diffsurv and Cox Partial Likelihood variants in isolation, we also run an additional experiment. For each method, we compute and time over 100 trials for a forward and backward pass on a NVIDIA GeForce GTX 1080 Ti using randomly generated logits. For diffsurv, this includes computing the predicted permutation matrix with differentiable sorting networks and applying the masking (Equation 15) and binary cross entropy (Equation 16). The possible permutation matrix generation is possible to precompute off GPU on the dataloader, so is not included in the timing. In Table 5, the Diffsurv methods, particularly Bitonic, consistently outperform CPL methods in compute time almost across all batch sizes and risk set sizes. As risk set size increases, CPL methods exhibit a decreasing trend in compute time, while Diffsurv's Odd-Even method experiences a notable rise, especially from risk set sizes of 32 to 128.

It is worth noting that CPL methods are currently computed over batches using a simple `for` loop, as the present implementations do not support batch parallel computation. However, there's potential for further optimization. For similar batch sizes and risk set sizes, we noted very similar overall convergence times with Diffsurv Bitonic variant being marginally faster than other methods. In the context of full model runs, the difference between Diffsurv and CPL in terms of training times is minimal; it is the model architecture that have a dominant effect on compute time.

Table 5: Isolated compute time for different methods, with various batch sizes and risk set sizes over 100 trials. Mean time and 95% confidence intervals are provided in milliseconds.

| Method | Batch Size, Risk Set Size | | | |
| --- | --- | --- | --- | --- |
| | 512, 2 | 128, 8 | 32, 32 | 8, 128 |
| Diffsurv: Odd-Even | $12.37 \pm 0.15$ | $25.63 \pm 0.09$ | $107.71 \pm 0.38$ | $305.91 \pm 3.66$ |
| Diffsurv: Bitonic | $3.85 \pm 0.03$ | $17.29 \pm 0.08$ | $55.49 \pm 0.47$ | $92.32 \pm 0.80$ |
| CPL: Breslow | $853.32 \pm 11.39$ | $281.22 \pm 1.20$ | $94.08 \pm 0.15$ | $25.45 \pm 0.46$ |
| CPL: Efron | $1729.87 \pm 5.54$ | $494.83 \pm 4.27$ | $164.73 \pm 1.40$ | $51.69 \pm 0.76$ |
| CPL: Ranked List | $719.42 \pm 4.99$ | $225.69 \pm 0.94$ | $74.36 \pm 0.18$ | $18.19 \pm 0.31$ |

## C Sorting Networks

There are multiple different types of sorting networks each with varying complexity. The ability to implement networks with the divide-and-conquer paradigm allows for sorting networks that scale more efficiently. Examples for Odd-Even and Bitonic sorting networks with $n = 8$ are shown in Figure 4. The latter allows construction of networks with size complexity $\mathcal{O}(nlog^2n)$ verses the $\mathcal{O}(n^2)$ in Odd-Even networks.

It is worth emphasising these are not neural networks. They are called "networks" because they are typically represented as diagrams that show how the items are compared and swapped as they are being sorted. Differentiable sorting networks do not introduce any additional parameters that need to be updated during optimization.

## D Calibration of Predicted Permutations

Model calibration in survival analysis models is essential for ensuring that the predicted probabilities of outcomes align closely with the true probabilities. An improperly calibrated model may lead

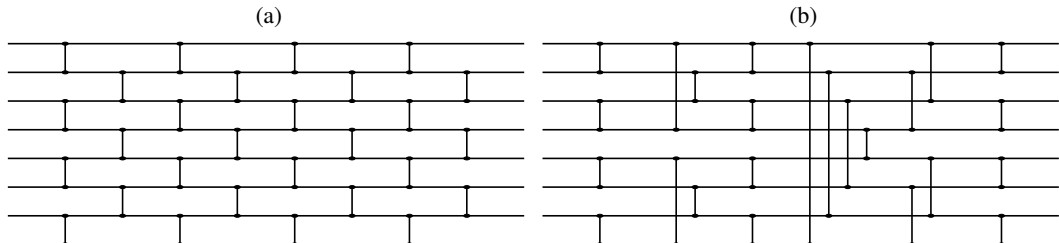

Figure 4: Example sorting networks of size 8; (a) Odd-Even, (b) Bitonic.

Table 6: Calibration of rank probabilities from predicted permutation matrices on survSVHN, keeping the number of events per batch equal. Mean (and standard deviation) Brier scores over 5 trails with different seeds. Corresponds with C-index results in Table 9.

| Method | Batch Size, Risk Set Size | | | |
|---|---|---|---|---|
| | 512, 2 | 128, 8 | 32, 32 | 8, 128 |
| CPL: Breslow | 0.081 (0.002) | 0.181 (0.001) | 0.056 (0.000) | 0.014 (0.000) |
| CPL: Efron | 0.081 (0.002) | 0.180 (0.001) | 0.055 (0.000) | 0.014 (0.000) |
| CPL: Ranked_List | 0.066 (0.001) | 0.185 (0.001) | 0.053 (0.000) | 0.012 (0.000) |
| Diffsurv: Bitonic | 0.059 (0.001) | 0.165 (0.001) | 0.043 (0.000) | 0.010 (0.000) |
| Diffsurv: Odd-Even | **0.056** (0.001) | **0.158** (0.001) | **0.039** (0.000) | **0.009** (0.000) |

556 to incorrect risk assessments and treatment decisions, potentially resulting in suboptimal patient
557 care and even adverse clinical consequences. We focus on the calibration of predicted individual
558 rankings, as demonstrated in Figure 5 and Table 6. Specifically, we qualitatively illustrate in Figure 5
559 that for a model with a risk set size of 8, both discrete predicted ranks and ranking probabilities are
560 accurately calibrated for the Diffsurv approach. To perform a quantitative comparison with baseline
561 methods, we need to derive ranking probabilities for the CPL model. Based on the assumptions
562 in Raykar et al. [2007], we assume that the probability of correct pairwise ordering for the CPL
563 adheres to the logistic function. We thus compute permutation matrices using differential sorting
564 networks, employing predicted partial log hazards as inputs and the logistic sigmoid function as the
565 differentiable sorting operator. By subsequently calculating Brier scores for the rank probabilities in
566 the predicted permutation matrices survSVHN, we analyze various combinations of batch size and
567 risk set size. Our findings show that the Diffsurv models consistently exhibit the lowest Brier scores
568 across all settings (refer to Table 6).

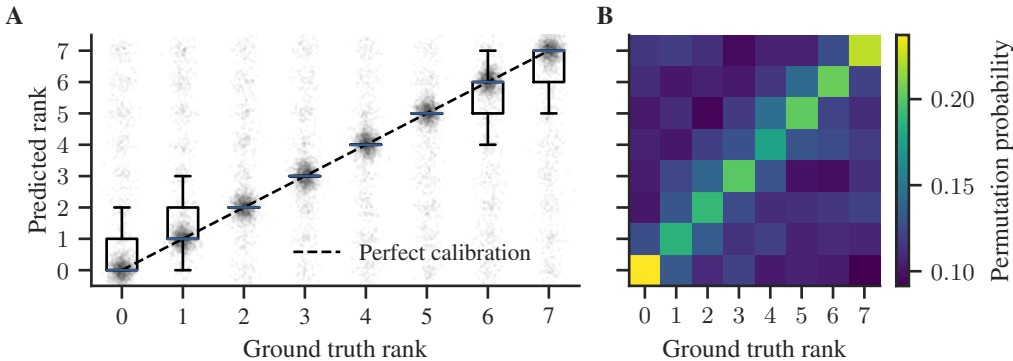

Figure 5: Calibration visualization of the Diffsurv (Bitonic) predicted ranking for a group of 8 subjects on the survSVHN dataset.

E **Effect of top-k loss variants on predictive performance**

The Top-K risk prediction variants for Diffsurv and the Cox Partial Likelihood, as introduced in Section 3.2, inherently bias the model to recognize only the top-k individuals with the highest risk. In this section, we explore the extent to which these specific variants influence the predictive performance of the models across the entire set of individuals. These results are shown in Table 7.

Table 7: C-index for real-world and semi-synthetic datasets for Top-k loss variants

| Top 10% prediction | FLCHAIN | NWTCO | SUPPORT | METABRIC | MIMIC IV CXR |
|---|---|---|---|---|---|
| Cox Partial Likelihood | .796 (.010) | .725 (.014) | .650 (.008) | .691 (.013) | .760 (.002) |
| CPL-TopK (Variant I) | .794 (.011) | .715 (.011) | .602 (.003) | .644 (.016) | .751 (.002) |
| CPL-TopK (Variant II) | .798 (.011) | .722 (.016) | .600 (.010) | .658 (.008) | .753 (.002) |
| Diffsurv | .794 (.011) | .696 (.025) | .634 (.008) | .649 (.016) | .761 (.002) |
| Diffsurv-TopK | .783 (.011) | .689 (.022) | .596 (.005) | .639 (.023) | .754 (.001) |

## F Additional Results

In Table 8 and Table 9, additional results for the MIMIC IV CXR and survSVHN imaging datasets are provided. Here, we maintain a constant total number of samples in each batch, which means that an increase in risk set size is compensated by a decrease in batch size. These results offer further understanding of the balance required between these two variables. We observed, while larger risk set sizes generally improve performance for both Diffsurv and CPH, the benefits tend to taper off as training can become more unstable and noisy with smaller batch sizes.

Table 8: Additional Results for MIMIC IV CXR. Mean and standard deviation of the C-index for different methods and batch risk set sizes. Bold indicates a significantly higher result with t-test and $p \leq 0.01$.[†] Most significant across all Batch Size, Risk Set Sizes.

| Method | Batch Size, Risk Set Size | | | |
|---|---|---|---|---|
| | 64, 2 | 4, 32 | 16, 8 | 1, 128 |
| Diffsurv: Bitonic | 0.761 (0.001) | **0.761** (0.000) | **0.763**[†] (0.001) | **0.761** (0.002) |
| Diffsurv: Odd-Even | 0.761 (0.002) | 0.756 (0.002) | 0.761 (0.001) | 0.749 (0.001) |
| CPL: Ranked List | 0.760 (0.002) | 0.755 (0.002) | 0.758 (0.003) | 0.755 (0.002) |

Table 9: Additional results for survSVNH keeping the number of events per batch equal. Mean (and standard deviation) over 5 trials with different seeds. Metric is C-index. Bold indicates a significantly higher result with t-test and $p \leq 0.01$.

| Method | Batch Size, Risk Set Size | | | |
|---|---|---|---|---|
| | 512, 2 | 128, 8 | 32, 32 | 8, 128 |
| Diffsurv: Odd-Even | **0.934** (0.001) | **0.940** (0.001) | **0.941** (0.001) | 0.933 (0.002) |
| Diffsurv: Bitonic | 0.931 (0.001) | 0.942 (0.001) | 0.940 (0.00166) | 0.928 (0.001) |
| CPL: Breslow | 0.905 (0.001) | 0.897 (0.001) | 0.910 (0.002) | 0.919 (0.001) |
| CPL Efron | 0.904 (0.002) | 0.898 (0.002) | 0.909 (0.003) | 0.918 (0.003) |
| CPL: Ranked List | 0.921 (0.001) | 0.922 (0.003) | 0.921 (0.001) | 0.917 (0.003) |