# OpenReview forum: "Differentiable sorting for censored time-to-event data."
_NeurIPS.cc/2023/Conference — NeurIPS 2023 poster_

### Official Review · Reviewer_e7KQ · 2023-07-05

**Soundness:** 3 good
**Presentation:** 3 good
**Contribution:** 3 good
**Rating:** 5
**Confidence:** 5

**Summary:**

This paper concerns a novel way to solve risk stratification for time-to-event data in the presence of right-censoring. Particularly, the authors solved the proportional hazard model via the differentiable sorting algorithm. Experiments were done on simulated and real-world benchmark datasets.

**Strengths:**

- The paper is well-written and easy to follow.
- The authors extended the differentiable sorting network to account for right-censoring by introducing a possible permutation matrix and applied it to survival analysis.
- This paper investigated the role of transitivity in the proportional hazards survival model.


**Weaknesses:**

- Neither the experimental nor theoretical advantage of Diffsurv over CPL is clear enough. I do not find a specific reason to use Diffsurv rather than CPL for training a PH survival model.

- Even if the authors showed some advantages of the proposed Diffsurv on the simulated dataset, it hardly shows a difference against CPL in real-world benchmark datasets.

- I believe Diffsurv is more than just one interesting way of solving the proportional hazards model. I think the differentiable sorting is expected to do better than the partial likelihood approach; however, the experimental results do not strongly support the advantage. The potential problem is the proportional hazard assumption - we observe marginal improvement as the PH assumption is too strong. Applying the differentiable sorting to models with no PH assumptions, such as CoxTime or DeepHit (as discussed in the appendix), may result in clear improvement (moreover, the experimental results also suggest that the method should be extended to non-PH because, under extremely simple simulated data that follows Weibull distribution with mild independent censoring, Diffsurv outperforms CPL but in real-world cases, the results hardly differ from those of the CPL’s suggesting that real-world applicability is questionable.).




**Questions:**

- I am not so sure about the clinical context of the top-$k$ risk stratification task. Could the authors kindly explain why the top-$k$ risk stratification is important enough to be evaluated separately, even if C-index accounts for the risk stratification?

- I do not think Kvamme et al. explicitly assumed pairwise independence, but they presented that it is sufficient to consider a risk set of size 1, which means having a risk set of size 1 is as good as having a larger risk set. Moreover, due to the nature of the random selection of SGD, the transitivity is implicitly considered. Also, in the paper, they investigated the impact of risk set size, which supports their claim “it is often sufficient to choose $n=1$.”

- I am curious how emphasizing top-K prediction changes the discrimination and calibration performance of survival models.

- Could authors kindly provide a computational cost analysis (either experimental or theoretical) of Diffsurv? It will be informative if the authors compare the computational costs of Diffsurv and CPL.

- RSF is not based on PH assumption. If the authors did not include DeepHit or other SOTA deep survival models just because they are not based on PH assumption, why is RSF given as a baseline?

**Limitations:**

The authors did not provide any theoretical analysis of Diffsurv. CPL might seem to be naive as it does not exhaustively consider all possible pairs, however, the estimator that maximizes the partial likelihood has nice statistical properties, including consistency and asymptotic normality. It would be nice if the authors could provide statistical/information theory behind Diffsurv and compare it with the existing ones.

---

> ### Author Rebuttal · Authors · 2023-08-09
>
> Thank you for your thorough review of our manuscript. We've addressed each point you raised and provided further clarifications where necessary:
>
> **Weakness 1 (Theoretical and practical advantages of the method)**
>
> *Theoretical Advantages*:
>
> Diffsurv integrates differentiable sorting networks, thereby introducing a transitive inductive bias into the survival ranking model. We have shown that this novel survival analysis approach performs competitively compared to CPL. More significantly, the ordering supervision enables straightforward implementation of new tasks, such as top-k risk prediction. In contrast, CPL cannot be readily adapted to predict permutation matrices directly. As we demonstrate in the global response and new experiments, attempts to adapt CPL for this purpose result in poorer calibration in the predicted rankings compared to Diffsurv.
>
> *Experimental Advantages*:
>
> Our experimental findings, detailed in Table 1 and lines L275-281, demonstrate Diffsurv outperforms CPL, particularly as transitivity increases within the data. This improvement is pronounced in larger risk set sizes, where our method leverages the inherent transitivity due to the introduction of sorting methods. Further experiments (Table 3) reveal consistent outperformance of CPL by Diffsurv in the top-k setting with significant clinical applications, like identifying top-k highest-risk individuals for specific treatment of preventative measures (see also our reply to Question 1 below).
>
> ---
>
> **Weakness 2 (Real-world benefits)**
>
> We acknowledge in the real-world risk stratification benchmark datasets, Diffsurv is only marginally better than the CPL baseline. We have introduced a novel approach to survival analysis using differentiable sorting algorithms that is at least competitive with established methods, but importantly also enables the development of novel downstream tasks using ordering supervision such as top-k risk prediction.
>
> ---
>
> **Weakness 3 (Potential of Diffsurv in non-PH settings)**
>
> We agree extending Diffsurv to time-dependent predictions without PH assumptions similar to CoxTime or DeepHit is a promising approach that will likely show a clearer performance benefit compared to the baselines and aim to address this in future work.
>
> ---
>
> **Question 1 (Significance of top-k risk prediction task)**
>
> We believe the top-k risk prediction task has important clinical applications: Whereas C-index accounts for risk stratification and measures the proportion of concordant pairs in all comparable pairs, the top-k risk prediction task as introduced in the manuscript tries to identify the set of top-k highest risk individuals, regardless of the orderings of individuals within the set of top-k highest individuals. The value of the top-k risk prediction task emerges in real-world clinical scenarios where resources like treatments or preventative interventions, such as vaccines, are constrained. In these situations, the primary concern is identifying high-risk individuals most likely to benefit from interventions, rather than predicting relative risk within that top tier. Risk ordering within the top-k individuals is secondary and may not affect immediate clinical decisions. Note this task is straightforward to implement with Diffsurv due to the predicted permutation matrix and thus showcases the advantages of including ordering supervision in a risk stratification model.
>
> ---
>
> **Question 2 (Pairwise independence and transitivity in Kvamme et al.)**
>
> We acknowledge that Kvamme et al. do not explicitly assume pairwise independence. However, it's worth noting optimisation with a risk set size of 1 inherently assumes pairwise independence. While Kvamme et al. demonstrate this is not an issue in their work, our empirical results contradict this. In our experiments, we observe benefits for larger risk sets, both for CPL and Diffsurv, though stronger for Diffsurv. The following differences in our study might explain this discrepancy:
>
> * We evaluate C-Index, while Kvamme uses MPLL, a lower bound of C-Index [1]
> * Our dataset and model complexity make optimisation more challenging, which might lead to our observed benefits for bigger risk sets
> * We have evaluated larger risk sets compared to Kvamme et al.'s study
>
> ---
>
> **Question 3: (Top-K prediction changes to discrimination and calibration)**
>
> While we did not include these results in the manuscript, we have noticed all model variants trained using top-k objectives perform slightly worse in risk stratification when evaluated using C-index.
>
> ---
>
> **Question 4 (Computational cost analysis)**
>
> We have now analysed the time complexity of the methods and run additional experiments to measure runtimes, please see the global response and the attached PDF.
>
> ---
>
> **Question 5 (Baselines)**
>
> We believe there is no stronger baseline than a tuned deep neural network with a CPL objective. While other models like DeepHit tend to outperform CPL in $\text{C-Index}^\text{td}$, for the ranking setting (non time-dependent) C-Index is the appropriate metric. We still include RSF as it is a commonly used baseline in the literature and is meaningfully different from Cox regression and neural networks trained with CPL.
>
> **Limitation 1 (Theoretical analysis of Diffsurv)**
>
> We agree CPL has some favourable statistical properties and appreciate the concern for a more thorough analysis of the theoretical properties of our proposed method. We would like to point out these align closely with the properties of sorting networks and differentiable sorting networks, which have been studied extensively in prior work [2, 3, 4].
>
> ---
>
> [1] Steck et al. 2007."On ranking in survival analysis: Bounds on the concordance index." NeurIPS.
>
> [2] Knuth, D. E. 1997. The Art of Computer Programming. Addison–Wesley.
>
> [3] Petersen et al. 2021. "Differentiable sorting networks for scalable sorting and ranking supervision." ICML.
>
> [4] Petersen et al. 2021. "Monotonic Differentiable Sorting Networks." ICML.

---

> > ### Author Response · Authors · 2023-08-16
> >
> > **Update on question 3**
> >
> > We now report additional experiments varying the Top K% and corresponding C-index performance. Please see the replies to Reviewer GX1f:
> > - https://openreview.net/forum?id=gYWjI7wLhc&noteId=9ucr4JN7g0
> > - https://openreview.net/forum?id=gYWjI7wLhc&noteId=TMw9rdHVPr

---

> > ### Comment · Reviewer_e7KQ · 2023-08-17
> >
> > I would like to thank the authors for their thorough feedback and additional experimental results. The authors addressed most of my concerns appropriately. However, still, I found some points are unclear to me.
> >
> > I have carefully gone through the discussion between the authors and reviewer GX1f regarding the top-k risk prediction task. I am still not fully convinced whether top-K risk prediction aligns with the conventional survival analysis scope. It might be better suited as a distinct task somewhat connected to survival analysis. However, with the C-index in each top-K group, it now seems like if there is a strong motivation to differentiate between high-risk and low-risk groups and to stratify the risk within the high-risk group, the concept of top-K risk prediction could potentially offer practical utility in the clinical field. Thus, Diffsurv having better Top-K accuracy as well as a better C-index in the Top-K group seems to demonstrate that Diffsurv has advantages over CPL.
> >
> > Nevertheless, I am still unsure of what advantages Diffsurv offers compared to CPL in the typical survival analysis context, especially when we limit our scope to the proportional hazards models. Even if the authors demonstrated that Diffsurv could show a better C-index than CPL as transitivity increases, such high transitivity may not be observed in real-world cases, as we see marginal improvement in C-index from the real-world dataset experiments.
> >
> > Overall, for me, Diffsurv looks more like a novel and promising ordering algorithm in the presence of censoring but not a complete survival analysis model. Also, the authors did not present a way to derive the survival function of Diffsurv, so it is not straightforward to evaluate the utility of Diffsurv as a survival analysis model. That is to say, additional investigation is still needed if the authors want to claim Diffsurv is for survival analysis. Therefore, I keep my score as is.
> >
> > One additional comment: If Diffsurv arranges the risks better than CPL, then Diffsurv may have better calibration performance than CPL (if the authors can derive survival probability from Diffsurv)

---

> > > ### Comment · Reviewer_e7KQ · 2023-08-17
> > > **Additional comment on the selection of the baselines**
> > >
> > > My comment on the baselines was not to encourage the authors to include more powerful or recent (deep) survival models. The authors said DeepHit is not included "since we do not model non-proportional hazards." My point was that even if RSF is not based on PH assumption it was included as a baseline. That was a bit awkward for me.

---

> > > ### Author Response · Authors · 2023-08-17
> > >
> > > We greatly appreciate your continued engagement and recognize the significance of the point you've highlighted regarding the positioning of Diffsurv within the domain of survival analysis.
> > >
> > > As we've delved into this topic, we too have grappled with the precise placement of Diffsurv within the broader landscape of survival analysis. This is reflected in our choice to emphasize "censored time-to-event data'' in the title. While it is possible to directly relate Diffsurv to existing survival methods, it is unique in its ability to directly predict permutation matrices, enabling novel tasks such as top-k identification.
> > >
> > > You are absolutely right in pointing out that Diffsurv currently does not produce traditional survival curves. This limitation is something we've acknowledged in our paper (L305). Merging the functionalities of Diffsurv with more traditional survival models to derive survival curves is an avenue we are actively exploring for future research. Nevertheless, we believe that the introduction of differentiable sorting and the novel capabilities introduced by Diffsurv are valuable in their own right.
> > >
> > > On reflection, we may term these capabilities as "survival ranking", as it aptly captures the essence of what Diffsurv does. However, we still believe that "survival analysis" serves as a broader and inclusive term, encompassing the subdomain of "survival ranking". This follows Raykar et al.’s introduction to the connections between the two domains (“we show that classical survival analysis involving censored data can naturally be cast as a ranking problem.”) [1]. We have updated the introduction and conclusion to make this distinction clearer.
> > >
> > > ---
> > >
> > > [1] Raykar et al. 2007. ‘On Ranking in Survival Analysis: Bounds on the Concordance Index’. NeurIPS.

---

> > > > ### Comment · Reviewer_e7KQ · 2023-08-17
> > > >
> > > > Considering that Diffsurv does not serve as a traditional survival model but rather focuses on survival ranking, which has potential utility in the clinical field, I agree that Diffsurv has clear advantages over CPL, as the new results demonstrate. I still expect some experimental investigations that are more focused on the traditional survival analysis perspective, but they can be left as a future study. I increase my score to borderline accept. Many thanks to the authors for their kind and sincere feedback.

---

### Official Review · Reviewer_Rdgo · 2023-07-05

**Soundness:** 3 good
**Presentation:** 3 good
**Contribution:** 3 good
**Rating:** 5
**Confidence:** 4

**Summary:**

The authors propose a deep learning-based survival model that utilizes a novel differentiable sorting objective capable of handling censored time-to-event data. In contrast to previous pairwise sorting objectives, the proposed listwise sorting objective achieves the transitive property inherent in survival data. Through semi-synthetic and real-world experiments, the authors demonstrate that the proposed method achieves comparable or improved risk ranking performance by leveraging the advantages of the proposed differentiable sorting objective.

**Strengths:**

Extending a differentiable sorting method to account for censoring and applying it to train survival models is novel.

**Weaknesses:**

-	The writing of the paper should be improved: it has many typos and grammatical errors. Some of the examples are listed below (note that there are many others not listed here):  Line 61: “extension differentiable” -> “extension of differentiable” / Line 97: “n=2” should be in mathematical form / Line 154: “It possible” -> “It is possible” / Line 161: “loss out we find” -> “loss, we find” / Line 77: “1-dimensional vector of size d” ->  “d-dimensional vector” / Make notations for referring to “Equation” consistent throughout the paper.

-	The motivation for using the “listwise” sorting method that achieves transitive property over the “pairwise” sorting method is not clearly stated throughout the paper (especially in the Introduction). It would be helpful to state the formal definition of transitive property in survival data with mathematical notations, and the limitations when this property is not maintained.

-	The design of semi-synthetic experiments does not elaborate why listwise sorting should outperform pairwise sorting.

-	The experiments are performed only with limited benchmarks (mainly the variants of partial likelihood methods and the proposed method itself) and evaluated only in terms of the discriminative power. The authors should provide predictive power (such as Brier scores) of survival models and in-depth qualitative analysis on the benefit of using the proposed listwise sorting method (when it has pros and when not).

-	It would be helpful if mathematical definitions are used for describing two scenarios of ranking right-censored samples in lines 187-191.


**Questions:**

-	Regarding Weakness 2, what is the limitation of pairwise sorting over listwise sorting? If correct pairwise sorting is achieved for all acceptable pairs, doesn’t it necessarily achieve transitive property?

-	Regarding Weakness 3, why listwise sorting over pairwise sorting is important in the semi-synthetic survSVHN?

-	Acceptable pairs do not consider cases in which the risk of a pair of samples cannot be directly compared. However, the proposed permutation ranking takes into account the possibility that right-censored samples may have higher ranks. The question arises: what is the benefit of including such potential cases for which we cannot guarantee a direct comparison? Furthermore, wouldn't it be harmful if a right-censored sample is highly likely to experience an early time-to-event, which can be inferred from the covariates? Can this be supported qualitatively based on the semi-synthetic (or synthetic) experiments?

-	In relation to Question 3, the semi-synthetic data generation process does not account for independent censoring, which assumes that the time-to-event and time-to-censoring are conditionally independent given the covariates. This assumption is commonly utilized in various survival analysis literature. Wouldn't the proposed semi-synthetic scenario be more favorable for utilizing the proposed permutation ranking method for sorting, rather than relying on acceptable pairs for sorting?



**Limitations:**

-	It seems that computing the Q matrix is computationally burdensome as it needs to compare all the sample pairs.

- Please see details in Weakness 2.

- Please see details in Weakness 4.

---

> ### Author Rebuttal · Authors · 2023-08-09
>
> We appreciate your feedback and hold your insights in high regard. Our goal is to respond to every concern you have mentioned:
>
> ---
>
> **Weakness 1 (Writing)**
>
> Thank you for the feedback on typographical and grammatical errors. We've addressed these and ensured greater consistency in notation and equation references.
>
> ---
>
> **Weakness 2 and Question 1 (Pairwise vs Listwise and Formal definition of transitivity)**
>
> The motivations for listwise over pairwise methods and their connection to transitivity warrant clarity. Recall that Cox Partial Likelihood (CPL) is a listwise approach, as indicated on L123. Let's highlight the benefits of listwise methods and then differentiate between differentiable sorting and CPL.
>
> Pairwise sorting methods handle ranking pairs individually. They establish local orderings (A over B, B over C) but may not ensure global consistency (A over C) throughout the set, as outlined in L113-115 and L266-274. Such limitations arise from focusing solely on local pairings. In some scenarios, these local orders can create inconsistencies.
>
> Increasing the minibatch size to include more pairs poses scalability challenges. Listwise methods evaluate comprehensive relationships among data points, scaling factorially. Ensuring pairwise order for every set may imply transitivity, but it's not always the case due to inherent inconsistencies when solely considering pairs. Refer to [1] for more on listwise motivations.
>
> Comparing CPL and Diffsurv, while both are listwise, they have distinct principles. CPL follows a top-one approach. Each term in the product ensures that one patient with an earlier observed event must higher rank than a subset of those observed later. Diffsurv instead considers the order of every patient in the set, and any incorrect ordering within each layer of the relaxed sorting network propagates to a higher predicted probability of permuting to an incorrect rank.
>
> ---
>
> **Weakness 3 and Question 2 (Transitivity in the semi-synthetic experiment)**
>
> It is important to highlight the role of transitivity in larger risk set sizes, which directly contributes to the higher performance of our method compared to the CPL baseline. The experiments shown in Table 1 and the details in lines L275-281 of the manuscript support this claim by demonstrating that both CPL and Diffsurv improve with larger risk set sizes. However, Diffsurv, due to the introduction of differentiable sorting methods, benefits more from the inherent transitivity in these larger risk sets, resulting in a more substantial performance delta.
>
> Furthermore, our results validate the significance of listwise sorting in the semi-synthetic survSVHN dataset. Table 2 provides an in-depth exploration of this, revealing that as the degree of transitivity within the data increases, Diffsurv consistently outperforms CPL. This divergence emphasises the importance of the differentiable sorting network in capturing the transitivity in the ranking task.
>
> ---
>
> **Weakness 4 (Benchmarking)**
>
> Our proposed method focuses on survival ranking, and we don’t learn a hazard function as in CPL, but rather a ranking function. Clinical decisions often hinge on relative comparisons, such as whether an observation affects survival time, rather than precise event timing. Additionally, there are specific applications in survival analysis where ranking is the primary objective, such as resource allocation among patients. In line with recent works in deep learning survival analysis, we chose CPL as the baseline and rigorously tuned the hyperparameters to ensure a fair comparison.
>
> Since our method is geared towards survival ranking, predicting event times and thus scores like IBS are not applicable in this situation. However, we recognise the value of assessing model calibration and appreciate your suggestion to include Brier scores for predictive power. We have incorporated the Brier scores for the permutation ranking in the attached PDF of our global response. Our findings demonstrate that Diffsurv outperforms the CPL baseline.
>
> ---
>
> **Weakness 5 (Mathematical notation)**
>
> We agree that adding the mathematical notation introduced in L73-77 and relating that to the possible ranks in L187-191 improves clarity and have adapted this section accordingly.
>
> ---
>
> **Question 3 (Ranking loss for right-censored samples)**
>
> In response to the concerns raised about the ranking of samples that cannot be directly compared, it is essential to highlight the design of our loss function, which has been constructed to solve this issue. The loss function (Equations 13 and 14) accounts for the potential challenges of incorporating right-censored samples. For an early right-censored sample $i$, $Q_{pi}$ is 1 almost everywhere, effectively sidestepping the risk comparison issue. The binary cross-entropy remains identical whether the model predicts uniform probability for all possible ranks or concentrates the probability mass on an early rank, inferred from the covariates. This ensures that the predicted rank falls within the possible ranks, considering censoring, without further influencing which rank within those possible ranks is preferable. We have now clarified this point in the methods section.
>
> ---
>
> **Question 4 (Censoring in semi-synthetic data)**
>
> To clarify, our semi-synthetic data generation uses only independent censoring. Is the question whether dependent censoring would favour our method in the benchmarks?
>
> ---
>
> **Limitation 1 (Runtime of calculating Qp)**
>
> We now report both theoretical time complexities of the differentiable sorting networks and empirical runtime results, including calculation of the $Q_p$ matrix in appendix section B.3 and show that the distinction in training times between Diffsurv and CPL is insignificant. Please also see our global response for further details on the runtime analyses.
>
> ---
>
> [1] Cao et al. 2007. ‘Learning to Rank: From Pairwise Approach to Listwise Approach’. ICML.

---

> > ### Comment · Reviewer_Rdgo · 2023-08-14
> > **Re: Rebuttal by Authors**
> >
> > I have read the authors' feedback (with additional experiments on the calibration performance) and updated my evaluation accordingly (from 4. Borderline reject to 5. Borderline accept). I appreciate the response to my comments but I still have concerns about the clinical utility of top-k predictions (which can be simply replaced with sorting based on individual risk predictions) and the absence of comparisons to more state-of-the-art deep learning methods.

---

> > > ### Author Response · Authors · 2023-08-14
> > > **Re: Re: Rebuttal by Authors**
> > >
> > > Thank you for taking the time to consider our rebuttal and adjusting your evaluation. We would like to shortly respond to the two remaining points:
> > >
> > > ---
> > >
> > > **Clinical Utility and Top-k Prediction**:
> > >
> > > The top-k prediction task has clinical utility in settings where a scarce resource, such as a new vaccine or a prevention checkup, must be allocated to the set of individuals most likely to benefit; usually, these are the individuals most at risk. In this setting, proper risk stratification is not necessary, as it is not required to correctly order individuals within the top-k most at risk. We only need to identify the individuals most at risk. We agree that, for a perfect risk model, one could identify the top-k most at risk by sorting based on individual risk predictions, as you suggested. However, we note that this is how the non-top-k baselines (e.g., the unadjusted Cox Partial Likelihood (CPL) model) in Table 3 are evaluated. Here, we simply train a standard CPL loss, sort individuals by predicted risk, and evaluate the top-k prediction performance. Table 3 also shows that, for real-world datasets with limited available data, the model variants suggested by us clearly outperform the unadjusted models, and that the Diffsurf-TopK model outperforms all other variants in this task.
> > >
> > > ---
> > >
> > > **State-of-the-art Deep Learning Baselines**:
> > >
> > > We want to emphasise that we do compare against extensively tuned deep learning baselines trained using several CPL loss variants, and we are unaware of any stronger time-independent baselines we could compare our method against. Are there specific state-of-the-art deep learning methods you had in mind?
> > >
> > > It's worth noting that directly comparing our method with time-dependent survival analysis methods, which model non-proportional hazards, presents challenges. Typically, these models need to be evaluated using metrics that are able to evaluate their time-dependant predictions, such as time-dependant c-index [1]. For instance, methods like DeepHit provide a probability mass function (PMF) for survival times, but the standard c-index necessitates ranking irrespective of time. This fundamental difference makes comparisons challenging (please note that the DeepHit paper also only evaluates the time-dependent C-index metric), and it is unlikely that these models improve upon Cox Partial Likelihood in the time-independent setting. We appreciate any insights or recommendations you might have on this matter.
> > >
> > > ---
> > >
> > > [1] Antolini, Laura, Patrizia Boracchi, and Elia Biganzoli. 2005. "A time‐dependent discrimination index for survival data.” Statistics in medicine.

---

### Official Review · Reviewer_RSfV · 2023-07-07

**Soundness:** 4 excellent
**Presentation:** 3 good
**Contribution:** 4 excellent
**Rating:** 7
**Confidence:** 3

**Summary:**

This paper investigates the problem of survival analysis under the proportional hazards (PH) hypothesis with deep learning models.

The main issue with previous approaches (Katzman et al. 2018 and Kvamme et al. 2019 as main representatives) is that the Cox loss is only computed in small mini-batches, potentially of size 2 in the case of the last reference. As a consequence, only pairwise ordering is imposed, potentially breaking transitivity.

This paper proposes to leverage recent advances in differentiable sorting to take into account transitivity in training deep survival models under the PH hypothesis. The main issue comes from censoring, which only implies a partial ordering. The main contribution of the authors is to propose a method that takes into account this censoring, defining a possible permutation matrix. The resulting method is flexible and can be adapted to plain survival or other tasks, such as top-K risk prediction.

Experiments on real data show that:
1. the method improves over previous deep learning methods (table 1), showing a +1 c-index point
2. the gap between both methods increases when the transitivity of the dataset increases (table 2)
3. this gap is maintained for other datasets (Table 3, first part) and other tasks (table 3, last part)

**Strengths:**

- Potential large impact: this paper provides a more sound way to deal with survival data and minibatches in deep learning, which could have a large impact
- Quality: the experiments are well conducted and yield convincing results
- Originality: to the best of my knowledge, this is one of the first papers leveraging differentiable sorting for survival analysis
- Clarity: the paper is well written, up to minor issues (see weaknesses)

**Weaknesses:**

- Clarity
  - notations in Equations (2), (3) and (4) are fuzzy. How is $j \in \mathcal{R} \ \lbrace i \rbrace$ chosen? It is unclear whether the sum applies to all such $j$ (as should be the case in eq. (4), since the c-index is computed over all acceptable pairs as defined L 106) or to only a random sample (as in Eq. (3)).

- Top-k task

  - As pointed out by the authors, due to the partial ordering the set $\mathcal{T}_k$ can grow very large (L217). As a consequence, the top-k-score defined in Eq (20) can be arbitrarily good. How do the authors control for this potential effect in the results?

**Questions:**

- cf top-k task: how do the authors control for the effect of the size of $\mathcal{T}_k$ on the top-k-score?
- Is there a difference in run-time between previous approaches and the proposed differentiable sorting?

**Limitations:**

- This paper is limited to proportional hazards assumptions, but it is clearly acknowledged by the authors

---

> ### Author Rebuttal · Authors · 2023-08-09
>
> We sincerely appreciate your thorough evaluation and positive feedback on our manuscript. Your insights into our work's potential impact and originality are highly encouraging. We acknowledge your concerns, especially regarding the clarity of notations and the handling of the top-k task. In this rebuttal, we strive to address each point further to enhance the quality and clarity of our contribution.
>
> ---
>
> **Weakness 1 (Clarity)**
>
> Here, we reproduce the notation from Kvamme et al. [1]. In theory, the sum applies to all such j as in the C-index. In practice, we must sample a limited number j’s according to the minibatch size. This is done randomly from the set of possible j’s.
>
> ---
>
> **Weakness 2 and Question 2 (Top-k task)**
>
> We thank the reviewer for highlighting the potential influence of the set size on the top-k-score due to partial ordering. Indeed, a large set size, stemming from partial ordering, can theoretically lead to an arbitrarily high top-k-score. However, a few points to note:
> * Our primary emphasis is on relative scores between models. While individual scores could be elevated due to partial orderings, the relative differences between models remain and can be used to assess their comparative performances.
> * Empirically, we observed that the scores manifest meaningful differences across methods and real-world datasets, including those with substantial censoring. These variations support the practical effectiveness of the metric.
>
> ---
>
> **Question 2 (Runtime analysis)**
>
> Thank you for the question on runtime differences between our proposed method and previous approaches, which we have now analysed in more detail in the global response and attached PDF.
>
> ---
>
> [1] Kvamme, Havard, Ørnulf Borgan, and Ida Scheel. "Time-to-Event Prediction with Neural Networks and Cox Regression." Journal of Machine Learning Research 20 (2019): 1-30.

---

> > ### Comment · Reviewer_RSfV · 2023-08-18
> > **Thank your for your rebuttal**
> >
> > Dear authors,
> >
> > Thank you for your answer and the additional experiments. I am satisfied regarding runtime and for the top-k task.
> >
> > Regarding clarity, thanks for the pointer to Kvamme et al. Although your notation matches the one of Equation (9) in Kvamme et al., I still think this is confusing for the reader: if a single point is sampled, I would recommend to a single index like $J(i)$:
> >
> > $L(\theta) = \prod_{i: \delta_i = 1} \frac{f_{\theta}(x_i)}{f_{\theta}(x_i) + f_{\theta}(x_{J(i)})},\text{ where }J(i) \sim \mathcal{R}\backslash \lbrace i \rbrace$
> >
> > If a random subset $\tilde{\mathcal{R}}\backslash \lbrace i \rbrace$ of size >1 is used, the notations of Eq (8) in Kvamme et al. are more precise.

---

> > > ### Author Response · Authors · 2023-08-19
> > >
> > > Thank you for your thoughtful feedback.
> > >
> > > We agree with your suggestions for clarity in this area, and in response, we have updated the manuscript to align with your recommendations.

---

### Official Review · Reviewer_GX1f · 2023-07-08

**Soundness:** 2 fair
**Presentation:** 2 fair
**Contribution:** 2 fair
**Rating:** 5
**Confidence:** 4

**Summary:**

The paper aims to address the survival problem as a ranking problem. The author asserts that by taking into account the transitivity property of ranking, sorting networks can be employed to tackle the current ranking problem effectively. Furthermore, the paper presents a solution to handle numerous potential permutation matrices in the presence of censoring. Additionally, it claims to outperform baseline methods in terms of top-k risk prediction. They tested their method on a semi-synthetic dataset based on the SVHN dataset and 5 real-world datasets.

**Strengths:**

By utilizing sorting networks, the paper leverages the inherent transitivity of ranking to its advantage. Additionally, it offers a comprehensive solution to handle all conceivable permutation matrices.

**Weaknesses:**

1. Survival analysis encompasses more than just generating rankings. In many cases, it is crucial to estimate the timing of the occurrence of the event of interest. The paper could address this aspect by comparing the learned hazard with the ground truth hazard in the semisynthetic experiment.
2. when it comes to top-k risk prediction, the method needs to provide a meaningful ordering, which, unfortunately, is not demonstrated in the paper.
3. It is not immediately apparent why this particular method is necessary. Is its sole purpose to accommodate the limitation of fitting the entire at-risk set into memory?
4. It would be beneficial for the author to provide a more detailed explanation of sorting networks and their functioning. The working principles of these networks are currently unclear to me. Furthermore, it would be helpful to know if sorting networks have any limitations regarding the size of the input list they can handle.
5. Improvements are not significant enough, even though baseline is a very simple method.


**Questions:**

1. What is the relationship between batch size and risk set size? How is it possible for the risk set size to be larger than the batch size?
2. I believe that as the risk set size in the Cox model increases, the performance of methods should be closer, but this does not seem to be the case (based on lines 111 and 112). Could you explain this to me?
3. Cox proportional hazards regression (CPL) forms the basis of Cox regression, and the hazard function in both methods should be the same. Consequently, the c-index should be the same for both methods. Why is it different?
4. According to the paper, the transitivity ratio should be between 0 and 1. However, it is reported as infinity. Why is that?


**Limitations:**

1. How does this method handle issues related to time-varying hazards?
2. As the method only provides risk scores, it is unclear how it can be assessed using metrics like IBS and Calibration.

---

> ### Author Rebuttal · Authors · 2023-08-09
>
> Thank you for your feedback. We value your insights and aim to address each concern you've raised. Here are our responses and clarifications:
>
> ---
>
> **Weakness 1 and Limitation 2 (Predicting the time of event vs ranking and calibration scores)**
>
> While estimating absolute timings for event occurrences is important in certain applications, many do not require this specific information. Decision-making through survival analysis frequently focuses on relative comparisons rather than absolute values. For instance, medical professionals may be more concerned with whether a particular observation will increase or decrease survival time rather than with the exact absolute value. Additionally, in some applications, the ranking itself is the primary objective of survival analysis. This may be evident in scenarios like assigning a limited discrete resource, such as a vaccine, to a subset of individuals most at risk.
>
> Importantly, we do not learn a hazard function as in CPL, but rather a ranking function. While it may be possible to recover a hazard function using adaptations of methods such as Breslow's estimator, we see this as future work.
>
> While we can’t assess commonly used calibration scores like IBS in the ranking setting, we now report Brier scores for the permutation ranking (see point model calibration in global response and attached PDF) and show that Diffsurv scores are better than the CPL baseline.
>
> ---
>
> **Weakness 2 (Meaningful ordering within top-k?)**
>
> The top-k task proposed in our paper intentionally does not focus on the order within the top-k, and we believe that this task has significant real-world clinical applications. For example, in scenarios where the objective is to identify the top-k highest-risk individuals for preventive measures (e.g., a vaccine with limited supply), the order within this group is irrelevant, as all selected individuals will be contacted. Therefore, we neither optimise for nor evaluate the meaningfulness of the predicted ordering of individuals within the top-k grouping. Instead, our results demonstrate that the models can accurately identify the top-k highest-risk individuals and that our method consistently outperforms the baselines in this context.
>
> ---
>
> **Weakness 3 (Necessity/motivation for proposed method)**
>
> Introducing differentiable sorting networks doesn't target the memory needs of large risk sets. By using these networks, our method integrates transitivity as an inductive bias into survival ranking, a novel approach to survival analysis with at least competitive results to established baselines. More importantly, Diffsurv establishes a framework that is also straightforward to build upon, such as in a top-k setting enabled by predicted permutation matrices from the differentiable sorting method. We note that established methods like CPL can’t easily be extended to predict permutation matrices directly, and we now show (see attached PDF in global response with new calibration experiments) that even if we extend CPL to do that, they exhibit worse calibration in predicted rankings compared to our method.
>
> ---
>
> **Weakness 4 (Explanation of sorting networks)**
>
> We have now expanded our description of sorting networks in Appendix E and added more references to traditional sorting networks (in addition to the papers introducing differentiable sorting networks).
>
> ---
>
> **Weakness 5 & Question 3 (Comparison with baselines)**
>
> We refer to Cox Partial Likelihood (CPL) as a loss function, without a specific covariate interaction, $h_\theta(x_i)$. Cox Regression refers to the scenario where linear interaction is maintained, while in CPL baselines, $h_\theta$ is parameterised with a neural network, explaining the better C-index performance. CPL is the default loss function in many recent deep learning survival analyses [1, 2]. We train our CPL baselines with the same architectures as Deepsurv, tuning the hyperparameters extensively. Other baselines, like DeepHit, are not expected to outperform CPL in this setting due to the non-proportional hazards affecting the time-dependent C-index but not the standard C-index. See our response to Weakness 1 regarding time dependency.
>
> ---
>
> **Question 1 (Risk set size vs Batch size)**
>
> We follow Petersen et al., 2021 [3] and train the model in batches of risk sets, i.e. each minibatch has shape (minibatch size, risk set size, feature size). Thus, for minibatch size 8 and risk set size 32, 256 samples are in the batch. This is described in more detail in section D.1 in the appendix.
>
> ---
>
> **Question 2 (Performance with increasing batch size)**
>
> The performance of both CPL and Diffsurv improves with larger risk set sizes, as shown in Table 1, but Diffsurv benefits more from larger risk set sizes. We believe this is because our method benefits more from the inherent transitivity in these larger risk sets due to the introduction of differentiable sorting methods. Please also see L275-281 in the manuscript for a more detailed explanation of this phenomenon and Table 2 with another experiment investigating this in more detail.
>
> ---
>
> **Question 4 (Transitivity ratio can’t be infinity?)**
>
> We don’t report a transitivity ratio of infinity. Instead, Table 2 shows that with an infinite number of quantiles (no discretisation as defined on L274), the transitivity ratio is .991.
>
> ---
>
> **Limitation 1 (Time-varying covariates)**
>
> We recognise the significance of models addressing time-varying hazards. While our method can be expanded in future work, our current focus is on fixed-time ranking and introducing differentiable sorting networks to risk analysis.
>
> ---
>
> [1] Buergel et al. 2022. ‘Metabolomic Profiles Predict Individual Multidisease Outcomes’. Nature Medicine.
>
> [2] Carr et al. 2021. ‘Longitudinal Patient Stratification of Electronic Health Records with Flexible Adjustment for Clinical Outcomes’. PMLR.
>
> [3] Petersen et al. 2022. Differentiable sorting networks for scalable sorting and ranking supervision. ICML

---

> > ### Comment · Reviewer_GX1f · 2023-08-15
> >
> > Having thoroughly reviewed both the author's response and the feedback from fellow reviewers, I've taken note of a notable observation regarding the c-index's decline in the context of top-k predictions, as highlighted in the response addressed to another reviewer. Given the acknowledged imperfections of the c-index as a metric, I believe it would be usefull to compare the c-index values between the data-generating process in the semi-synthetic experiment and the model. It is important that the c-index associated with the model does not surpass that of the data-generating process.
> > The concept of employing sorting methods is appealing; however, it's important to underscore that the entire concept is rooted in a score that isn't flawless (we recognize that the log-likelihood serves as the definitive score for survival analysis).
> > In the interest of comprehensive analysis, it would be advantageous to conduct a study exploring the impact of varying the 'k' parameter in top-k predictions. Such an investigation could shed light on how diverse methods perform as 'k' increases. The internal mechanics governing the ordering within the top-k framework remain somewhat opaque, and I'm eager to know whether it yields an acceptable c-index within the top-k subset.
> > Taking into account the insights gleaned from the author's response, I am inclined to adjust my evaluation to lean towards a borderline rejection.

---

> > > ### Author Response · Authors · 2023-08-16
> > >
> > >
> > > Thank you for taking the time to consider our rebuttals, in particular noting the innovative use of differentiable sorting within survival analysis. We will address your remaining points and sincerely hope that our clarifications will further persuade you towards recommending acceptance of our paper.
> > >
> > > ---
> > >
> > > **C-index concerns**
> > >
> > > We agree that the C-index performance of the model should not exceed that of the data generation process. As such, we have calculated the C-index of the data generation process using the ground truth risk values and observed sampled times, and found a value of 0.980. This is significantly higher than the best performing model (Diffsurv) with a score of 0.943. This difference underscores the inherent challenge of the task, and even with a sophisticated model like Diffsurv, there remains room for improvement.
> > >
> > > The C-index is widely recognized as the de-facto metric in clinical practice for evaluating survival models, and most machine learning papers in survival analysis use variants of the C-index as the primary evaluation metric as well. While we acknowledge that the C-index, like any metric, has its limitations and trade-offs, its predominant use by practitioners as the primary evaluation metric for survival analysis indicates that optimising models for this metric can have a significant impact. More theoretically, Raykar et al. [1] summarises why we think that treating survival analysis as a ranking problem is reasonable:
> > > > In this paper, we show that classical survival analysis involving censored data can naturally be cast as a ranking problem. The concordance index (CI), which quantifies the quality of rankings, is the standard performance measure for model assessment in survival analysis. In contrast, the standard approach to learning the popular proportional hazard (PH) model is based on Cox’s partial likelihood. We devise two bounds on CI–one of which emerges directly from the properties of PH models–and optimise them directly.
> > >
> > > [1] Raykar et al. 2007. ‘On Ranking in Survival Analysis: Bounds on the Concordance Index’. NeurIPS.
> > >
> > > ---
> > >
> > > **Top-k evaluation**
> > >
> > > We acknowledge that evaluating the top-k prediction task for $k \neq 10$ will further underscore the robustness of our results, and have now evaluated all model variants for varying k on the tabular real-world data sets. As you can see from the table below, the results are largely consistent across datasets and k. For NWTCO, the top-25 metric is not informative due to the very high rate of censoring in this dataset.
> > >
> > > Note: Metric is Top K% where K is indicated in the column.
> > >
> > > **FLCHAIN**
> > > | Model | Top 5% | Top 10% | Top 25% |
> > > |:-|:-:|:-:|:-:|
> > > | Cox Partial Likelihood | .314 (.012) | .462 (.017) | .699 (.029) |
> > > | CPL-TopK (Variant I) | .375 (.041) | .468 (.015) | .818 (.031) |
> > > | CPL-TopK (Variant II) | .369 (.040) | .465 (.008) | .717 (.026) |
> > > | Diffsurv | .326 (.038) | .468 (.015) | .709 (.039) |
> > > | Diffsurv-TopK | .388 (.023) | .488 (.016) | .825 (.036) |
> > >
> > > **SUPPORT**
> > > | Model | Top 5% | Top 10% | Top 25% |
> > > |:-|:-:|:-:|:-:|
> > > | Cox Partial Likelihood | .255 (.027) | .286 (.014) | .403 (.014) |
> > > | CPL-TopK (Variant I) | .499 (.050) | .481 (.016) | .520 (.008) |
> > > | CPL-TopK (Variant II) | .489 (.043) | .475 (.018) | .523 (.012) |
> > > | Diffsurv | .255 (.047) | .304 (.027) | .409 (.019) |
> > > | Diffsurv-TopK | .553 (.049) | .521 (.023) | .560 (.009) |
> > >
> > > **METABRIC**
> > > | Model | Top 5% | Top 10% | Top 25% |
> > > |:-|:-:|:-:|:-:|
> > > | Cox Partial Likelihood | .179 (.063) | .247 (.054) | .534 (.046) |
> > > | CPL-TopK (Variant I) | .627 (.161) | .507 (.120) | .587 (.028) |
> > > | CPL-TopK (Variant II) | .640 (.137) | .500 (.063) | .589 (.046) |
> > > | Diffsurv | .263 (.061) | .325 (.064) | .555 (.028) |
> > > | Diffsurv-TopK | .587 (.165) | .547 (.113) | .643 (.020) |
> > >
> > > **NWTCO**
> > > | Model | Top 5% | Top 10% | Top 25% |
> > > |:-|:-:|:-:|:-:|
> > > | Cox Partial Likelihood | .311 (.100) | .400 (.070) | 1.000 (.000) |
> > > | CPL-TopK (Variant I) | .379 (.117) | .418 (.067) | 1.000 (.000) |
> > > | CPL-TopK (Variant II) | .358 (.109) | .416 (.055) | 1.000 (.000) |
> > > | Diffsurv | .337 (.082) | .390 (.067) | 1.000 (.000) |
> > > | Diffsurv-TopK | .384 (.124) | .416 (.056) | 1.000 (.000) |
> > >
> > > *Continued in the next comment...*

---

> > > > ### Author Response · Authors · 2023-08-16
> > > >
> > > > *Continued from above*
> > > >
> > > > ---
> > > >
> > > > Top-k prediction, by design, does not optimise for the order within the top-k. Nonetheless, we agree that it is important to understand the relationship between identifying the top-k individuals and the order within this group. In the top-k setting, a higher risk translates to a higher probability to be in the top-k individuals, and thus higher risk individuals should have a higher predicted rank. The following table shows the C-Index within the predicted top-k individuals for the tabular datasets. We find that ordering within this top-k group is still reasonable for top-k specific methods when compared to baselines.
> > > >
> > > > Note: Metric is C-index within top K% where K is indicated in the column.
> > > >
> > > > **FLCHAIN**
> > > > | Model | Top 5% | Top 10% | Top 25% |
> > > > |:-|:-:|:-:|:-:|
> > > > | Cox Partial Likelihood | .641 (.056) | .646 (.028) | .671 (.016) |
> > > > | CPL-TopK (Variant I) | .613 (.045) | .649 (.025) | .672 (.006) |
> > > > | CPL-TopK (Variant II) | .615 (.046) | .652 (.029) | .666 (.017) |
> > > > | Diffsurv | .572 (.046) | .606 (.030) | .652 (.013) |
> > > > | Diffsurv-TopK | .607 (.065) | .635 (.039) | .668 (.014) |
> > > >
> > > > **SUPPORT**
> > > > | Model | Top 5% | Top 10% | Top 25% |
> > > > |:-|:-:|:-:|:-:|
> > > > | Cox Partial Likelihood | .583 (.036) | .588 (.016) | .593 (.017) |
> > > > | CPL-TopK (Variant I) | .666 (.018) | .663 (.022) | .624 (.016) |
> > > > | CPL-TopK (Variant II) | .666 (.027) | .671 (.014) | .640 (.015) |
> > > > | Diffsurv | .587 (.043) | .607 (.007) | .567 (.010) |
> > > > | Diffsurv-TopK | .663 (.022) | .615 (.029) | .558 (.034) |
> > > >
> > > > **METABRIC**
> > > > | Model | Top 5% | Top 10% | Top 25% |
> > > > |:-|:-:|:-:|:-:|
> > > > | Cox Partial Likelihood | .597 (.072) | .584 (.059) | .604 (.025) |
> > > > | CPL-TopK (Variant I) | .670 (.072) | .676 (.066) | .625 (.049) |
> > > > | CPL-TopK (Variant II) | .696 (.082) | .689 (.048) | .631 (.024) |
> > > > | Diffsurv | .613 (.087) | .553 (.079) | .526 (.030) |
> > > > | Diffsurv-TopK | .618 (.105) | .594 (.085) | .552 (.029) |
> > > >
> > > > **NWTCO**
> > > > | Model | Top 5% | Top 10% | Top 25% |
> > > > |:-|:-:|:-:|:-:|
> > > > | Cox Partial Likelihood | .630 (.038) | .653 (.061) | .686 (.034) |
> > > > | CPL-TopK (Variant I) | .672 (.091) | .662 (.041) | .677 (.040) |
> > > > | CPL-TopK (Variant II) | .651 (.049) | .642 (.035) | .701 (.032) |
> > > > | Diffsurv | .618 (.062) | .629 (.037) | .668 (.035) |
> > > > | Diffsurv-TopK | .628 (.076) | .643 (.061) | .623 (.090) |
> > > >
> > > > ---
> > > >
> > > >
> > > > We recognize the value of including an upper bound C-index from the data-generated ground truth and further Top-K experiments. These will be integrated into the camera-ready, and we thank you again for your suggestions.

---

> > > > > ### Comment · Reviewer_GX1f · 2023-08-16
> > > > >
> > > > > considering the fact that authors have responded to my questions and the fact that ranking networks can be a promising line of research for survival analysis I increase the score to borderline accept.

---

### Author Rebuttal · Authors · 2023-08-09

We sincerely appreciate the thorough and constructive feedback provided by all four reviewers.
As suggested by multiple reviewers, we have further investigated the runtime (both theoretically and empirically) and calibration of our proposed survival ranking method.

---

**Theoretical and empirical analysis of runtime**

Diffsurv and the baseline models using CPL use the same neural network architectures. The operations relating to these networks are the main bottleneck for all methods, particularly for the imaging datasets. Only minor differences in runtime are due to the respective Cox Partial Likelihood (CPL) and differentiable sorting operations.

*Theoretical Time Complexities*: First, we highlight the number of operations required for the differentiable sorting networks: Odd-Even as $\mathcal{O}(n^2)$ and Bitonic sorting networks as $\mathcal{O}(nlog^2n)$ as discussed in Appendix C and [1]. The time complexity of our primary baseline, CPL, is $\mathcal{O}(nlogn)$ as it also requires sorting the event times.

*Benchmarking Process*: We have conducted experiments to understand these differences (see results in the attached PDF). We measured the time taken for a forward and backward pass on an NVIDIA GeForce GTX 1080 Ti, utilising randomly generated logits. This experiment provides a more isolated measure of the compute times, allowing for a focused comparison between the methods. For Diffsurv, our timing includes the predicted permutation matrix computation via differentiable sorting networks and the subsequent masking (Equation 13) and binary cross-entropy (Equation 14). Importantly, we precomputed the potential permutation matrix ($Q_{p}$) generation off-GPU (same as during minibatch training), so it isn't included in this benchmark.

*Results*: As documented in Table 1, our Diffsurv methods, especially Bitonic, have a noticeable edge over CPL methods regarding compute time across various batch and risk set sizes. It's interesting to observe that as the risk set size goes up, CPL methods show a reduced compute time, while Diffsurv's Odd-Even variant sees a substantial increase, particularly when moving from risk set sizes of 32 to 128.

*Optimization Possibilities*: It's crucial to note that current implementations of CPL methods compute over batches using a straightforward 'for' loop since they don't support batch-parallel computation. There's significant room for further optimisation here.

*Overall Observations*: In comprehensive model runs, the difference in training times between Diffsurv and CPL is insignificant. The compute time is primarily driven by the model architecture. Specifically, we observed similar convergence times across methods, with the Diffsurv Bitonic variant being slightly quicker.

We recognise the importance of computational efficiency in practical deployments and point to [1] for further analysis of differentiable sorting network compute time.

---

**Model calibration**

It is important to note that we do not learn a hazard function as in CPL, but rather a ranking function, and thus can’t assess predictive model calibration using commonly used scores like IBS. We acknowledge that model calibration in survival analysis models is essential for ensuring that the predicted probabilities of outcomes align closely with the true probabilities.

We include a new analysis (see attached PDF), focussing on the calibration of predicted individual rankings. Specifically, we first qualitatively illustrate in Figure 1 in the attached PDF that discrete predicted ranks and ranking probabilities are accurately calibrated for a model with a small risk set size for the Diffsurv approach. To perform a quantitative comparison with baseline methods, we need to derive ranking probabilities for the CPL model. Based on prior work [2], we assume that the probability of correct pairwise ordering for the CPL adheres to the logistic function. We thus compute permutation matrices using differential sorting networks, employing predicted partial log hazards from a pretrained model as inputs and the logistic sigmoid function as the differentiable sorting operator. By subsequently calculating Brier scores for the rank probabilities in the predicted permutation matrices and ground truth permutations derived from the hazards in the survSVHN dataset, we analyse various combinations of batch size and risk set size. Our findings show that the Diffsurv models consistently exhibit the lowest Brier scores across all settings (Table 2 in the attached PDF).

---

[1] Petersen, F., Borgelt, C., Kuehne, H., & Deussen, O. (2021, July). Differentiable sorting networks for scalable sorting and ranking supervision. In International Conference on Machine Learning (pp. 8546-8555). PMLR.

[2] Steck, Harald, et al. "On ranking in survival analysis: Bounds on the concordance index." Advances in neural information processing systems 20 (2007).

---

> ### Author Response · Authors · 2023-08-21
> **Summary of Discussion Period**
>
> As we approach the end of the discussion period, we would like to take a moment to summarise:
>
> - **Clarifications:** We have offered in-depth responses to the raised concerns. Notably, discussions centered on Diffsurv's placement within the survival analysis domain and the clinical utility of both the ranking and Top-K tasks.
>
> - **Additional Experiments:** Prompted by valuable feedback, we have conducted further experiments. These offer insights into the Top-K task, especially concerning varying K values and alterations in C-index performance.
>
> We truly appreciate the constructive feedback and insights from all reviewers during the discussion. Coupled with the initial rebuttal, this dialogue has undeniably elevated our work's quality. We remain open to any concluding remarks or suggestions as the review period wraps up.

---

### Decision · Program_Chairs · 2023-09-21

**Decision:**

Accept (poster)

**Comment:**

This paper proposes the use of differentiable sorting networks to rank right censored patients by risk (for the problem of survival analysis). This is achieved via the construction of a possible permutation matrix. The matrix captures whether its feasible for a point i to permute to rank j (and if so its set to 1) and otherwise set to 0. The numbers are setup to respect the ranking of patients in the censored and event set. The models is trained using a sorting network that enables models to learn to sort risks. The model is trained on the surSVHN dataset and the as well as standard benchmarks in survival analysis. The method appears to do reasonably well compared to learning with the Cox proportional likelihood based method when evaluated on c-index.

Overall I find this is a modest paper that I lean towards accepting because it provides a self-contained approach to ranking right censored data that sets itself apart from traditional methods for ranking. While I do think it has differences (both pros and cons) vs traditional methods for survival analysis I strongly recommend the authors make this transparent to readers at the outset (e.g. this method produces no survival curves). In addition, my understanding based on this work (though this is never explicitly stated) is that this model is expected to rank data when the outcome process is censored (conditionall) at random (given the covariates) -- it is worth making this assumption explicit so that practitioners are aware of the censorship domain this model was intended to work well in.